# Essential Paralogous Proteins as Potential Antibiotic Multitargets in *Escherichia coli*

Christine D. Hardy[a]*

[a]CDH Consulting, Irvine, California, USA

**ABSTRACT** Antimicrobial resistance threatens our current standards of care for the treatment and prevention of infectious disease. Antibiotics that have multiple targets have a lower propensity for the development of antibiotic resistance than those that have single targets and therefore represent an important tool in the fight against antimicrobial resistance. In this work, groups of essential paralogous proteins were identified in the important Gram-negative pathogen *Escherichia coli* that could represent novel targets for multitargeting antibiotics. These groups include targets from a broad range of essential macromolecular and biosynthetic pathways, including cell wall synthesis, membrane biogenesis, transcription, translation, DNA replication, fatty acid biosynthesis, and riboflavin and isoprenoid biosynthesis. Importantly, three groups of clinically validated antibiotic multitargets were identified using this method: the two subunits of the essential topoisomerases, DNA gyrase and topoisomerase IV, and one pair of penicillin-binding proteins. An additional eighteen protein groups represent potentially novel multitargets that could be explored in drug discovery efforts aimed at developing compounds having multiple targets in *E. coli* and other bacterial pathogens.

**IMPORTANCE** Many types of bacteria have gained resistance to existing antibiotics used in medicine today. Therefore, new antibiotics with novel mechanisms must continue to be developed. One tool to prevent the development of antibiotic resistance is for a single drug to target multiple processes in a bacterium so that more than one change must arise for resistance to develop. The work described here provides a comprehensive search for proteins in the bacterium *Escherichia coli* that could be targets for such multitargeting antibiotics. Several groups of proteins that are already targets of clinically used antibiotics were identified, indicating that this approach can uncover clinically relevant antibiotic targets. In addition, eighteen currently unexploited groups of proteins were identified, representing new multitargets that could be explored in antibiotic research and development.

**KEYWORDS** antibiotic resistance, antibiotic targets, multitargeting

Antibiotic resistance in bacterial pathogens is an ongoing problem, with an estimated 1.2 million deaths worldwide caused by antibiotic-resistant bacteria in 2019 (1). As bacteria develop resistance to existing antibiotics, the discovery and development of new antimicrobial compounds is necessary to avoid a return to unacceptable pre-antibiotic era-levels of infectious disease mortality (2). Drugs having novel targets or mechanisms are particularly desirable due to the lack of preexisting resistance to such agents. The aim of this work is to inform the prioritization of novel antibiotic targets by identifying potential multiprotein targets, which are less prone to developing high-level drug resistance than single targets (3, 4).

An ideal antibiotic target has several general characteristics. First, it is essential for bacterial viability, being part of a cellular component or biosynthetic pathway required for cell growth, cell division, and/or maintenance of cellular integrity. Second, it is present in a range of bacteria, having considerable homology at the drug-binding site in the spectrum of bacteria to be targeted. Third, the target gene product does not have significant similarity, at least in the drug-binding region, to human proteins, thus allowing for selectivity of bacterial killing over

Address correspondence to christinedhardy@gmail.com.

*Present address: Christine D. Hardy, Department of Molecular Biology and Biochemistry, University of California at Irvine, Irvine, California, USA.

The author declares no conflict of interest.

host toxicity. Finally, the drug does not readily elicit resistance, which would render it ineffective after a short period of use.

The emergence of antimicrobial drug resistance can occur by several mechanisms, including mutation of the target gene, which can occur rapidly for drugs that target a single gene product (3, 4). One approach to slow this type of drug resistance is for a molecule to target essential gene products or structures encoded by multiple genes, a concept known as multitargeting (5). Target-related resistance to a multitarget drug requires that all involved genes mutate, making the development of high-level resistance much slower and less likely. Indeed, most clinically used systemic antibiotics have multiple ligands (3, 6), including the quinolone antibiotics, which target two essential topoisomerases in bacteria (DNA gyrase and topoisomerase IV) and the $\beta$-lactam antibiotics, which target multiple peptidoglycan synthesis enzymes (the penicillin-binding proteins, PBPs). Nonprotein multitargets such as rRNA, the cellular membrane, and cell wall components are other important multitargets exploited by clinically used antibiotics (4).

It is important to note that some forms of antimicrobial resistance are not prevented by multitargeting. In particular, nontarget-related mutations that alter the permeability of the Gram-negative outer membrane or lead to an increase in drug efflux can lead to clinically relevant drug resistance (7). Other nontarget-related mechanisms of resistance include inactivation of the antibiotic itself, as exemplified by the $\beta$-lactam-hydrolyzing $\beta$-lactamases, functions often encoded on mobile elements that may confer multiple resistance phenotypes (8). Antibiotic resistance may also be mediated by the presence or acquisition of an alternative, drug-resistant variant of an antibiotic target, such as the $\beta$-lactam-resistant PBP MecA, characteristic of methicillin-resistant *Staphylococcus aureus* (7), or by the horizontal transmission of target protection mechanisms, such as rRNA methylases (5, 7). Despite these alternative pathways to resistance, multitargeting is a key attribute of most successful systemic antibiotics, likely because it prevents the development of target-related high-level single-step resistance observed at high frequencies for single-target agents.

In this work, I conducted a comprehensive genomic search to identify groups of proteins that could be used as multitargets in the bacterium *Escherichia coli*, a defining member of the important family of Gram-negative bacteria, the *Enterobacteriaceae*. Resistance to antimicrobials in pathogenic organisms of this family causes significant mortality and health care burden, and the discovery of novel agents to treat drug-resistant *Enterobacteriaceae* is considered a top priority by the World Health Organization (9). *E. coli* itself was responsible for more antibiotic-resistance-associated deaths than any other bacterial pathogen in a recent worldwide study (1).

Generally, multitargeting of protein targets requires a degree of sequence homology at the amino acid level between at least two essential targets. For example, the quinolone targets, DNA gyrase (gyrase) and topoisomerase IV (topo IV), share a high level of protein sequence homology, as do the essential *E. coli* penicillin-binding proteins, PBP2 and PBP3. Proteins within an organism that share sequence homology are called paralogs. Paralogous essential proteins carry out independent roles, each of which is essential for cell viability, yet the similarity between the proteins can allow for targeting of multiple proteins with a single-agent antibiotic.

Although families of paralogous proteins were noted soon after the publication of the first bacterial genomes (10) and essential genes have been defined in many bacteria, a genomic-scale description of paralogous essential proteins that could be investigated for antibiotic multitargeting has not been reported. To this end, I carried out automated BLAST searches on each protein sequence from a representative pathogenic *E. coli* genome, allowing for exploration of the paralogous protein landscape across *E. coli* strains. These data were parsed to identify all essential gene products having at least one essential *E. coli*-conserved paralog, creating a genomic-scale list of potential protein antibiotic multitargets in *E. coli*.

Using this approach, 21 groups of *E. coli*-conserved essential paralogous proteins were identified. Of these, three protein groups were identified that are existing targets of clinically used multitargeting antibiotics: the two subunits of gyrase and topo IV, as well as the penicillin-binding proteins MrdA (PBP2) and FtsI (PBP3), indicating that important

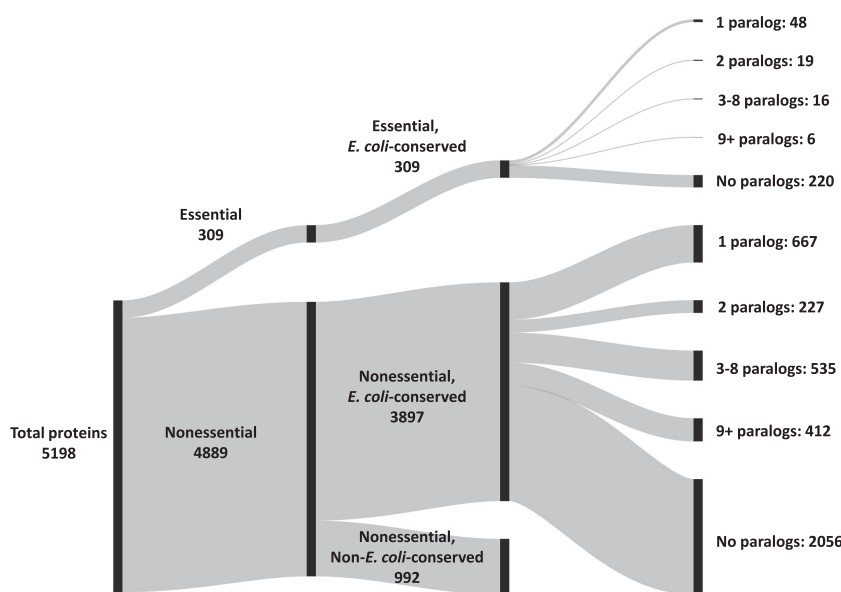

**FIG 1** Summary of the paralog analysis of the *E. coli* O157:H7 str. Sakai genome. Protein sequences were first sorted by whether they are essential or nonessential, then by whether they were found to be conserved within *E. coli*, and finally by how many *E. coli*-conserved paralogs were counted. The number of proteins in each category is indicated.

drug targets can be uncovered using this method. The additional 18 groups comprise proteins that are not currently multitargets of approved antibiotics and represent a potential starting point for drug discovery efforts aimed at the development of novel multitargeting antibiotics.

## RESULTS

**Genomic-scale identification of paralogous proteins in *E. coli*.** For this study, the well-annotated pathogenic *E. coli* O157:H7 strain Sakai genome was used as the source for protein sequences. Each protein-coding gene product (5,198 total, see Table S1) from the *E. coli* Sakai genome was passed through a Python script to conduct a BLAST search of the amino acid sequence against all *E. coli* genomes. These data were compiled to create a table listing the number of genomes having one or more sequence matches (E value, ≤0.001) for each protein (Table S3). In general, the first match corresponds to the exact or near-exact sequence being found, while additional matches represent paralogous sequences. To be considered *E. coli* conserved, a sequence match was required to be present in at least 90% of the calculated number of *E. coli* genomes queried, ensuring generality of the results across *E. coli* strains. The list of essential genes used here (Table S2) was compiled based on previous studies (see Materials and Methods).

A summary of the results from the paralog analysis is presented in Fig. 1. All 309 essential proteins were found to be conserved within *E. coli*, as were 79.7% (3,897) of proteins marked nonessential. A total of 89 essential proteins had one or more *E. coli*-conserved paralogous sequence, representing 28.8% of essential proteins. By comparison, 1,841 (47.2%) of nonessential *E. coli*-conserved proteins had at least one conserved paralog. Of the 89 essential proteins with at least one *E. coli*-conserved paralog, 48 proteins had 1 conserved paralogous sequence (representing 15.5% of all essential proteins), 19 had 2 conserved paralogous sequences (6.1% of essential proteins), 16 had 3 to 8 conserved paralogous sequences (5.2% of essential proteins), and 6 had 9 or more conserved paralogous sequences (1.9% of essential proteins).

Within the group of 89 essential proteins having at least 1 conserved paralog, 44 proteins had matches to at least 1 additional essential gene product, while 40 had only nonessential paralogs (Table S4). In addition, 5 essential proteins (CydC, MsbA, LolD, LptB, and FtsE) had 54 or more paralogous matches in *E. coli* Sakai. These proteins are from the ABC transporter

family and were not evaluated further, as this is a large protein family present in all organisms, including humans.

The 44 essential proteins with *E. coli*-conserved essential paralogs were classified by cross-correlation into 21 protein groups that represent potentially promising multitargets for antibiotic development. Each of these proteins was subjected to additional BLAST searches to assess the relative strength of the matches (by E value) to the *E. coli* paralog(s) versus potential matches to human proteins and was evaluated for conservation in other bacteria. Additional information considered for each protein included COG (Clusters of Orthologous Genes) functional category, cellular localization, the presence of enzymatic activity, the existence of described inhibitors, and the availability of protein structural information. These data are summarized in Table 1.

Each group of essential paralogous proteins is discussed in detail below, starting first with clinically validated multitarget protein families, then with nonexploited multitargets lacking human homologs, and finally with nonexploited multitargets having human homologs.

**Groups 1 to 3: clinically validated multiprotein targets.** The two protein groups (groups 1 and 2) with the highest degree of homology (lowest E values) between the paralogs contain the subunits of the bacterial toposimerases, gyrase and topo IV. These enzymes are responsible for untangling DNA during DNA replication and maintaining supercoiling homeostasis of the bacterial chromosome (11). Gyrase is composed of the two subunits GyrA and GyrB, and topo IV is composed of ParC and ParE. GyrA and ParC are paralogs, as are GyrB and ParE. All four gene products are essential.

The discovery of the GyrA/ParC and GyrB/ParE groups in this study provides an important validation for this method of multiprotein target discovery. Indeed, these enzymes are a remarkable pair in their very high level of subunit homology, their high degree of conservation in bacteria, and the presence of multiple enzymatic activities available for inhibition. The clinically important quinolone class of antibiotics (e.g., ciprofloxacin, levofloxacin, etc.) bind to the GyrA and ParC subunits of gyrase and topo IV and arrest the enzymes in the middle of their catalytic cycle. Novel compounds targeting gyrase and topo IV using different mechanisms are in clinical development (12–15), further solidifying the importance of these targets in the antibacterial space.

One group of penicillin-binding proteins was also identified in this study (group 3), composed of the FtsI (PBP3) and MrdA (PBP2) proteins. FtsI and MrdA are essential peptidoglycan transpeptidases and are important targets for $\beta$-lactam drugs (16). $\beta$-Lactams are among the oldest antibiotics and remain an extremely important tool in medicine today.

$\beta$-Lactam antibiotics target PBPs by inhibiting their transpeptidase activity, which is responsible for cross-linking peptidoglycan strands. Peptidoglycan, a polysaccharide matrix cross-linked with pentapeptides, is the major component of the bacterial cell wall and is required for structural integrity and maintenance of cell shape in most bacteria (17). Peptidoglycan must be synthesized during cell elongation and cell division, making the enzymes involved in this process powerful intervention points for inhibiting bacterial cell propagation. Nonessential PBPs, including those encoded by MrcA, MrcB, PbpC, and MtgA (forming a single group) and DacA, DacC, DacD, and PbpG (forming another group), were also identified in this study as paralogous groups but are not discussed further, as this work focused on essential paralogous gene products.

The generation of known antibiotic targets in groups 1 to 3 indicates that this method of paralogous essential protein search can yield clinically relevant targets for multitargeting therapeutics. The following sections describe additional groups generated using this method that do not have existing multitarget inhibitors in clinical use and could represent promising new targets for antibiotic development. These protein groups are broadly divided into two sets: groups 4 to 9, which do not have human homologs, and groups 10 to 17, which have human homologs.

**Groups 4 to 9: potential novel multiprotein targets without human homologs.** This study identified six protein groups that could represent the most promising candidates for multitarget antibiotics in that the proteins in these groups have no human homologs.

The first of these (group 4) comprises the FtsW and RodA (also called MrdB) proteins, both of which are present in the inner membrane of *E. coli*. These essential proteins are well

**TABLE 1** Groups of essential, *E. coli*-conserved paralogous proteins identified in this study[a]

| Group no. | Protein name | Protein description | Essential paralog(s) | Region of homology to essential paralog(s) (aa numbering) | E value of match to essential paralog(s) | COG functional category[b] | Cellular localization | Closest human homolog | Region of homology to closest human homolog (aa numbering) | E value of match to closest human homolog | Conservation in bacteria[b] Gi(+) | Gi(−) | Atyp | Mt | Cd | Enzymatic activity | Known inhibitors[c] | Representative PDB structures[d] | Nonessential paralog(s) |
|---|---|---|---|---|---|---|---|---|---|---|---|---|---|---|---|---|---|---|---|
| 1 | GyrA | DNA gyrase, subunit A | ParC | 1–743 | 6e-126 | L | Cytosol | DNA topoisomerase II alpha | 32–371 | 2e-14 | + | + | + | + | + | DNA cleavage/reunion | Quinolones, gepotidacin (12), other NBTIs (15) | 4CKK, 6RKU, 6RKV, 3NUH | None |
| 1 | ParC | DNA topoisomerase IV, subunit A | GyrA | 1–680 | 5e-126 | L | Cytosol, IM | DNA topoisomerase II beta | 9–197 | 1e-08 | + | + | + | − | − | DNA cleavage/reunion | Quinolones, gepotidacin (12), other NBTIs (15) | 1ZVU, 7LHZ, 5EIX | None |
| 2 | GyrB | DNA gyrase, subunit B | ParE | 1–550, 735–794 | 2e-125, 3e-06 | L | Cytosol | DNA topoisomerase II alpha | 25–543 | 5e-30 | + | + | + | + | + | ATPase | Coumarins, SPR720 (13), other ATP-site inhibitors (112), zoliflodacin (14) | 4WUB, 6RKU, 6RKV, 3NUH | None |
| 2 | ParE | DNA topoisomerase IV, subunit B | GyrB | 1–542, 559–621 | 5e-122, 2e-06 | L | Cytosol | DNA topoisomerase II beta | 65–619 | 3e-30 | + | + | + | − | − | ATPase | Coumarins, SPR720 (13), other ATP-site inhibitors (112), zoliflodacin (14) | 1S16, 7LHZ, 5EIX | None |
| 3 | FtsI | Peptidoglycan transpeptidase (*E. coli* PBP3) | MrdA | 15–574 | 1e-33 | D, M | IM | None | NA | NA | + | +[e] | +/− | + | + | Peptidoglycan D, D-transpeptidase | β-lactams, ETX0462 (82) | 4BJP, 7JWL | None |
| 3 | MrdA | Peptidoglycan transpeptidase (*E. coli* PBP2) | FtsI | 13–617 | 2e-33 | D, M | IM, periplasm | None | NA | NA | +/− | +[e] | +/− | + | + | Peptidoglycan D, D-transpeptidase | β-lactams, ETX0462 (82) | 6G9P | None |
| 4 | FtsW | Cell division peptidoglycan glycosyltransferase | RodA (MrdB) | 91–403 | 7e-43 | D, M | IM | None | NA | NA | + | + | +/− | + | + | Peptidoglycan glycosyltransferase | None | 6BAR, 6PL5, 6PL6 | None |
| 4 | RodA (MrdB) | Cell elongation peptidoglycan glycosyltransferase | FtsW | 55–365 | 2e-46 | D, M | IM | None | NA | NA | +/− | + | +/− | + | + | Peptidoglycan glycosyltransferase | None | 6BAR, 6PL5, 6PL6 | None |
| 5 | LolC | Lipoprotein release complex subunit | LolE | 4–396 | 4e-35 | M | IM, periplasm | None | NA | NA | +/− | +/− | − | − | − | None | G0507 (23), SMT-738 (24, 22), other preclinical (21, 22) | 5NAA, 7MDX, 7MDY, 7ARI | None |
| 5 | LolE | Lipoprotein release complex subunit | LolC | 2–409 | 1e-34 | M | IM | None | NA | NA | +[f] | +/− | +/− | − | − | None | G0507 (23), SMT-738 (24), other preclinical (21, 22) | 7MDX, 7MDY, 7ARI | None |
| 6 | RpoD | RNA polymerase major σ70 subunit | RpoH | 375–599 | 2e-18 | K | Cytosol | None | NA | NA | + | + | + | + | + | None | None | 6XL5 | RpoS, FliA |
| 6 | RpoH | RNA polymerase heat shock σ32 subunit | RpoD | 49–279 | 3e-20 | K | IM, cytosol | None | NA | NA | + | − | − | − | − | None | None | None | RpoS |
| 7 | DnaA | DNA replication initiator protein | Hda | 119–364 | 3e-16 | L | IM, cytosol | None | NA | NA | + | + | + | + | + | ATPase | None | 2EOG, 1J1V, 3R8F, 2Z4R | None |
| 7 | Hda | Inhibitor of reinitiation of DNA replication | DnaA | 17–247 | 2e-16 | L | IM, cytosol | None | NA | NA | − | − | − | − | − | None | None | 5X06, 3BOS | None |
| 8 | LpxA | Catalyzes the first reaction of lipid A biosynthesis | LpxD | 12–210 | 1e-09 | M | Cytosol | None | NA | NA | + | +/− | +/− | − | − | UDP-N-acetylglucosamine acyltransferase | Various preclinical (31–33, 35, 36) | 1LXA, 2QIA | None |
| 8 | LpxD | Catalyzes the third reaction of lipid A biosynthesis | LpxA | 122–334 | 1e-09 | M | Cytosol | None | NA | NA | + | +/− | +/− | − | − | UDP-3-O-(3-hydroxymyristoyl) glucosamine N-acetyltransferase | Various preclinical (34–36) | 3EH0, 4IHF, 6P8B, 3PMO | None |
| 9 | MurD | Catalyzes the addition of the first amino acid in peptidoglycan monomer | MurC | 74–333 | 1e-06 | M | Cytosol | None | NA | NA | + | + | +/− | + | + | UDP-N-acetylmuramoyl-alanine ligase | Various preclinical (38–42, 45) | 2F00, 1P3D, 1P31 | Mpl |
| 9 | MurC[g] | Catalyzes the addition of the second amino acid in peptidoglycan monomer | MurD | 79–309 | 1e-04 | M | Cytosol | None | NA | NA | + | + | +/− | + | + | UDP-N-acetylmuramoyl-L-alanine:D-glutamate ligase | Various preclinical (43–45) | 1UAG, 2Y66, 2Y1O | None |
| 10 | PrfA | Peptide release factor RFI | PrfB | 10–346 | 7e-66 | J | Cytosol | Mitochondrial translational release factor 1-like | 59–346 | 1e-88 | + | + | + | + | + | Hydrolysis of peptidyl-tRNA when associated with ribosome | Apidaecins (47–49), preclinical (50) | 5J3C, 5O2R, 1RQ0 | PrfH |
| 10 | PrfB | Peptide release factor RFII | PrfA | 28–362 | 9e-64 | J | Cytosol | Mitochondrial translational release factor 1-like | 60–359 | 4e-51 | + | +/− | +/− | + | + | Hydrolysis of peptidyl-tRNA when associated with ribosome | Apidaecins (47), preclinical (50) | 1GQE, 6OG7, 5MDV | PrfH |
| 11 | Ffh | Signal recognition particle protein component | FtsY | 41–299 | 3e-40 | U | Cytosol | Signal recognition particle 54-kD protein (SRP54) | 4–448 | 3e-62 | + | + | + | + | + | GTPase | Goadsporin (51) | 7O9I, 2XXA | None |
| 11 | FtsY | Signal recognition particle receptor | Ffh | 196–498 | 1e-42 | U | IM, cytosol | Signal recognition particle 54-kD protein (SRP54) | 228–498 | 1e-34 | + | + | + | + | + | GTPase | Preclinical fragments (52) | 2YH5, 7O9H, 2XXA | None |
| 12 | IspA | Catalyzes the first and second steps in polyisoprenoid biosynthesis | IspB | 43–290 | 9e-21 | H | Cytosol | All trans-polyprenyl-diphosphate synthase (PDSS1) | 21–258 | 5e-14 | + | +/− | +/− | + | + | Farnesyl diphosphate synthase | Bisphosphonates (55) | 1RTR, 1RQJ | None |

**TABLE 1** (Continued)

| Group no. | Protein name | Protein description | Essential paralog(s) | Region of homology to essential paralog(s) (aa numbering) | E value of match to essential paralog(s) | COG functional category[b] | Cellular localization | Closest human homolog | Region of homology to closest human homolog (aa numbering) | E value of match to closest human homolog | G(+) | G(−) | Atyp | Mt | Cd | Enzymatic activity | Known inhibitors[c] | Representative PDB structures[d] | Nonessential paralog(s) |
|---|---|---|---|---|---|---|---|---|---|---|---|---|---|---|---|---|---|---|---|
| 12 | IspB | Catalyzes reactions forming the isoprenoid chain of ubiquinone-8 and menaquinone-8 | IspA | 43–275 | 8e-16 | H | Cytosol | All trans-polyprenyl-diphosphate synthase (PDSS1) | 32–323 | 2e-42 | + | + | +/− | + | − | Octaprenyl diphosphate synthase | Bisphosphonates (56) | 3WJK, 5ZHE | None |
| 13 | DnaX | DNA polymerase III clamp loader γ and τ subunits | HolB | 35–170 | 2e-09 | L | Cytosol | Replication factor C, subunit 5 | 9–313 | 2e-14 | + | + | + | + | + | ATPase | None | 1NJF, 1NJG, 1JR3, 3GLF | RarA |
| 13 | HolB | DNA polymerase III clamp loader δ' subunit | DnaX | 21–159 | 4e-14 | L | Cytosol | Replication factor C, subunit 5 | 108–210 | 3e-04 | + | + | + | + | + | None | None | 1A5T, 1JR3, 3GLF | None |
| 14 | Der (EngA) | Ribosome biogenesis GTPase | Era | 5–154, 205–385 | 6e09, 7e-08 | J | Cytosol | GTP-binding protein 3 (mitochondrial) | 155–370, 5–91 | 6e-10, 1e-04 | + | + | + | + | + | GTPase | Preclinical (62, 63) | 5DN8, 3J8G | MnmE (TrmE) |
| 14 | Era | Ribosome biogenesis GTPase | Der (EngA) | 11–165, 11–186 | 4e09, 4e-08 | J | IM, cytosol | GTPase Era (mitochondrial) | 11–281 | 2e-20 | + | + | +/− | + | + | GTPase | Preclinical (63) | 1EGA, 3IEU | MnmE (TrmE) |
| 15 | RibD | Catalyzes the second and third steps of riboflavin biosynthesis | TadA | 4–158 | 5e-11 | H | Cytosol | None | NA | NA | + | +/− | +/− | + | + | Diaminohydroxy phosphoribosyl aminopyrimidine deaminase/5-amino-6-(5-phosphoribosylamino) uracil reductase | None | 2G6V, 8DQB | None |
| 15 | TadA | tRNA adenosine deaminase | RibD | 9–167 | 2e-08 | J | Cytosol | tRNA adenosine deaminase 2 | 9–150 | 1e-23 | + | + | +/− | + | + | Deamination of adenosine to inosine at position 34 of tRNA$^{Arg2}$ | None | 1Z3A | None |
| 16 | TsaB | Posttranscriptional modification of tRNAs | TsaD | 1–92 | 1e-07 | J | Cytosol | None | NA | NA | + | + | + | + | + | None | None | 4YDU, 6Z81, 3ZEU | None |
| 16 | TsaD | Posttranscriptional modification of tRNAs | TsaB | 1–109 | 2e-07 | J | Cytosol | O-sialoglycoprotein endopeptidase-like 1 (OSGEPL1) | 3–331 | 4e-55 | + | + | + | + | + | Transfer of threonylcarbamyl (TC) from TC-AMP to A$^{37}$ of substrate tRNAs | None | 4YDU, 6Z81, 3ZEU | None |
| 17 | FabI | Catalyzes a key regulatory step in fatty acid biosynthesis | FabG | 6–251 | 7e-07 | I | Cytosol | L-xylulose reductase | 3–251 | 1e-13 | + | + | +/− | + | − | Enoyl-ACP reductase | Triclosan, isoniazid, afabicin (67), CG-549 (68), MUT056399 (69), preclinical (70-73) | 1QSG, 1DFI, 4CV2, 5CFZ | HdhA, UcpA, YghA, BdcA |
| 17 | FabG | Catalyzes the first reductase step of each cycle of fatty acid biosynthesis | FabI | 5–241 | 7e-07 | I | Cytosol | 3-oxoacyl-ACP reductase | 6–244 | 2e-58 | + | + | +/− | + | + | 3-oxoacyl-ACP reductase | Preclinical (70-73) | 1Q7B, 6TSX, 6T77 | 17 Nonessential oxidoreductases |
| 18 | ValS | Valine-tRNA ligase | IleS, LeuS | IleS:30–762; LeuS: 1–355, 420–766 | IleS: 2e-36; LeuS: 4e-27, 3e-03 | J | Cytosol | Valine-tRNA ligase | 1–932 | 0.0 | + | + | + | + | + | Valine-tRNA ligase | None | 1GAX | None |
| 18 | IleS | Isoleucine-tRNA ligase | ValS, LeuS, MetG | ValS: 46–775; LeuS: 58–629; MetG: 58–115 | ValS: 3e-34; LeuS: 4e-12; MetG: 2e-05 | J | Cytosol | Isoleucine-tRNA ligase, mitochondrial | 4–930 | 0.0 | + | + | + | + | + | Isoleucine-tRNA ligase | Mupirocin, CB-432 (114), SB-203207 and SB-203208 (115) | 1QU2 | None |
| 18 | LeuS | Leucine-tRNA ligase | ValS, IleS, MetG | ValS: 1–379, 417–777; IleS: 42–646; MetG: 33–181 | ValS: 2e-25, 2e-03; IleS: 4e-12; MetG: 3e-06 | J | Cytosol | Probable leucine-tRNA ligase, mitochondrial | 29–859 | 2e-166 | + | + | + | + | + | Leucine-tRNA ligase | Epetraborole (76), GSK656 (77), agrocin (116) | 4ARC, 4AS1 | None |
| 18 | MetG | Methionine-tRNA ligase | LeuS, IleS | LeuS:6–151; IleS: 15–72 | LeuS: 2e-06; IleS,1e-05 | J | Cytosol | Methionine-tRNA ligase, cytoplasmic | 6–543 | 2e-70 | + | + | + | + | + | Methionine-tRNA ligase | CRS3123 (74), REP8839 (75), preclinical (117) | 1F4L, 6SPO, 6WQS, 6WQT | YgiH |
| 19 | LysS | Lysine-tRNA ligase | AspS, AsnS | AspS: 61–345, 356–499; AsnS: 63–499 | AspS: 2e-19, 5e-07; AsnS: 7e-10 | J | Cytosol | Lysine-tRNA ligase | 13–503 | 3e-130 | + | + | + | + | + | Lysine-tRNA ligase | None | 1BBU, 1BBW | LysU, EpmA |
| 19 | AspS | Aspartate-tRNA ligase | LysS, AsnS | LysS: 11–298, 415–555; AsnS: 454–563, 14–259 | LysS: 3e-19, 5e-07; AsnS: 2e-09, 4e-05 | J | Cytosol | Aspartate-tRNA ligase, mitochondrial | 1–589 | 2e-147 | + | + | + | + | + | Aspartate-tRNA ligase | Microcin C (116) | 1C0A, 1EQR | LysU, EpmA |
| 19 | AsnS | Asparagine-tRNA ligase | LysS, AspS | LysS: 15–458; AspS: 360–466, 16–277 | LysS: 6e-10; AspS: 1e-09, 3e-05 | J | Cytosol | Probable asparagine-tRNA ligase, mitochondrial | 6–464 | 2e-124 | +/− | + | +/− | − | + | Asparagine-tRNA ligase | None | 6PQH, 1X54 | LysU |
| 20 | GltX | Glutamate-tRNA ligase | GlnS | 3–124 | 5e-11 | J | Cytosol | Probable glutamate-tRNA ligase, mitochondrial | 1–462 | 1e-66 | + | + | + | + | + | Glutamate-tRNA ligase | None | 7K86, 4G6Z | GluQ |

**TABLE 1** (Continued)

| Group no. | Protein name | Protein description | Essential paralog(s) | Region of homology to essential paralog(s) (aa numbering) | E value of match to essential paralog(s) | COG functional category[h] | Cellular localization | Closest human homolog | Region of homology to closest human homolog (aa numbering) | E value of match to closest human homolog | Conservation in bacteria[b] | | | | | Enzymatic activity | Known inhibitors[c] | Representative PDB structures[d] | Nonessential paralog(s) |
|---|---|---|---|---|---|---|---|---|---|---|---|---|---|---|---|---|---|---|---|
| | | | | | | | | | | | G(+) | G(−) | Atyp | Mt | Cd | | | | |
| 20 | GlnS | Glutamine-tRNA ligase | GltX | 28–149 | 6e-11 | J | Cytosol | Glutamine-tRNA ligase | 28–552 | 4e-147 | + | − | +/− | − | + | Glutamine-tRNA ligase | None | 1OOB, 1QTQ | None |
| 21 | ProS | Proline-tRNA ligase | ThrS | 6–199, 404–571 | 4e-08, 4e-06 | J | Cytosol | Probable proline-tRNA ligase, mitochondrial | 10–234, 373–567 | 4e-63, 2e-22 | + | + | + | + | + | Proline-tRNA ligase | None | 5UCM | None |
| 21 | ThrS | Threonine-tRNA ligase ProS | 232–422, 473–633 | 5e-08, 4e-06 | | J | Cytosol | Threonyl-tRNA synthetase | 4–642 | 4e-152 | + | + | + | + | + | Threonine-tRNA ligase | Borrelidin (118), obafluorin (119), other preclinical (120) | 1QF6, 1EVK, 1EVL | None |

<sup></sup>

[a]G(+), Gram-positive bacteria; G(−), Gram-negative bacteria; Atyp, atypical bacteria; *Mt, Mycobacterium tuberculosis; Cd, Clostridioides difficile*; IM, inner membrane; OM, outer membrane; NBTI, novel bacterial topoisomerase inhibitor; aa, amino acid; NA, not applicable.

[b]See Table S5 for detailed information about the bacterial conservation analysis.

[c]Inhibitors in bold underlined text are clinically approved; those in plain text are or have been in clinical trials; those in underlined text are at the preclinical stage.

[d]Protein structures in the Protein Data Bank (PDB) from *E. coli* or from bacterial homologs having an amino acid sequence with ≥50% identity are listed in regular text; structures from bacterial homologs having <50% homology are listed in italics. See Table S6 for additional information about available protein structures and references for PDB entries.

[e]PBPs from different organisms have different naming systems (121). See Table S5 for which PBPs were considered to be homologous.

[f]Some organisms have a single protein, annotated as LolCE or simply LolE, that shares similarity to both LolC and LolE (122). Here, these were considered to be LolE homologs.

[g]MurD was counted as having no paralogs in the initial automated BLAST search used in this step.

[h]COG functional categories are as follows: L, replication, recombination, and repair; D, cell cycle control, cell division, chromosome partitioning; M, cell wall/membrane/envelope biogenesis; K, transcription; J, translation; U, intracellular trafficking, secretion, and vesicular transport; H, coenzyme transport and metabolism; I, lipid transport and metabolism; ribosomal structure and biogenesis; U, coenzyme transport and metabolism.

conserved across bacteria and share a high degree of sequence homology. Initially annotated as lipid flippases, the functions of FtsW and RodA have recently been more fully elucidated: both proteins have now been shown to possess peptidoglycan glycosyltransferase activity (18–20). Peptidoglycan synthesis requires both glycosyltransferase activity, to grow the glycan chains, and transpeptidase activity (carried out by PBPs), to cross-link them. RodA and FtsW act in concert with the PBPs MrdA and FtsI, respectively, to effect peptidoglycan synthesis during cell elongation and cell division. To date, no inhibitors of FtsW or RodA have been described. The strong similarity between FtsW and RodA, their involvement in the validated peptidoglycan biosynthetic pathway, and the presence of a targetable enzymatic activity strongly signal that these bacterial-specific proteins represent an important new dual target for antibiotic development.

The next paralogous protein group is composed of the proteins LolC and LolE (group 5). These essential proteins are present in the inner membrane of Gram-negative bacteria and are part of the LolCDE lipoprotein release complex. In combination with an outer membrane component, the LolCDE complex transports lipoproteins from the inner to the outer cell membrane. Preclinical inhibitors of the LolCDE complex have been described (21–24). The exact mode of inhibition by these compounds is unclear, but resistance to most of them is readily achieved with single point mutations, indicating that they are not targeting multiple sites. The presence of the LolCDE complex in the cytoplasmic membrane, the extensive degree of homology between LolC and LolE, and the absence of similar proteins in humans make these proteins attractive targets for dual-targeting compounds for use against some Gram-negative bacteria.

Group 6 comprises four bacterial sigma factors: RpoD, RpoH, RpoS, and FliA. Sigma factors bind to the core RNA polymerase complex and to DNA, targeting the transcription machinery to specific promoter sequences. RpoD encodes the primary sigma factor, $\sigma^{70}$, while RpoH encodes $\sigma^{32}$, the heat shock response sigma factor. Both RpoD and RpoH are essential. RpoS and FliA are nonessential and encode the stress response sigma factor, $\sigma^{S}$, and the flagellar synthesis-specific sigma factor, $\sigma^{28}$, respectively.

Targeting of bacterial transcription by antibiotics is the mechanism for the rifamycin class of antibiotics as well as the newer antibiotic, fidaxomicin (25). These compounds inhibit the activity of the core RNA polymerase enzyme, responsible for DNA-templated RNA synthesis. Inhibitors targeting proteins outside the core polymerase have been explored (26) but have not been developed into clinical candidates. While sigma factors do not have enzymatic activity on their own, their important interaction with RNA polymerase, their broad conservation in bacteria, and their lack of a closely related human homolog may make them suitable multitargets for antibiotic discovery.

Group 7 is composed of the DNA-binding protein DnaA and its regulator, Hda. DnaA binds to and opens the bacterial origin of DNA replication, recruiting the replication machinery to initiate replication of the bacterial chromosome. Following initiation, Hda stimulates DnaA to hydrolyze its bound ATP, preventing reinitiation of DNA synthesis. Both proteins are essential for viability in *E. coli*, although Hda is not as broadly conserved in bacteria as DnaA (27). Given their opposing roles in DNA replication, it is possible that partially inhibiting both proteins would counteract the effects of inhibiting each individually. Nevertheless, mistimed DNA replication can clearly lead to bacterial cell death (28), and inhibition of multiple proteins involved in the initiation of DNA synthesis could lead to complex lethal effects.

Group 8 contains the lipid A biosynthetic pathway members, LpxA and LpxD. Lipid A forms the membrane anchor for lipopolysaccharide (LPS), an essential component of the outer membrane in Gram-negative bacteria. Another member of this pathway, LpxC, is a well-studied antibiotic target, with two inhibitors having reached clinical trials (29, 30). Inhibitors for both LpxA (31–33) and LpxD (34), individually, have been designed, and dual-targeting LpxA/LpxD small molecules (35) and peptide inhibitors (36) have also been described. Though necessarily restricted to Gram-negative bacteria, continued effort into dual targeting of this validated pathway could be a promising research avenue.

The MurC and MurD enzymes of group 9 are involved in the early cytoplasmic phase of peptidoglycan biosynthesis in which the monomeric unit of peptidogylcan is

formed. The first committed step in this pathway is catalyzed by MurA, which is the target of the clinically used antibiotic fosfomycin (37). MurC and MurD have been subjected to extensive small-molecule inhibitor screens (38), and several pre-clinical-stage inhibitors of MurC and MurD individually have been identified (38–45). In addition, weak dual-targeting inhibitors of MurC and MurD have been described (45). Given the importance of peptidoglycan synthesis as an antibiotic target, further work to identify stronger dual inhibitors of these enzymes may be worthwhile.

**Groups 10 to 21: potential novel multiprotein targets with human homologs.** An additional 12 protein groups contain potentially promising paralogous proteins for targeting with antibiotics, but one or several essential members of each group also has at least one human homolog. It should be noted that human homologs are present for many clinically important antibiotic targets, including the bacterial topoisomerases, but there is enough divergence between the human and bacterial enzymes that bacterial-specific inhibitors have been successfully developed.

Group 10 is composed of peptide release factor I (PrfA) and peptide release factor II (PrfB), which bind to the ribosome in the presence of an mRNA stop codon, facilitating release of the newly synthesized polypeptide. The ribosome is a well-known target of many antibiotic classes (e.g., aminoglycosides, tetracyclines, chloramphenicol, macrolides, etc.) (46), but translation termination has not been clinically utilized as an antibiotic target to date (47).

PrfA and PrfB are intriguing targets, as both proteins are essential for translation termination, having nonoverlapping stop codon specificity. PrfA and PrfB are targets of a class of insect-produced antimicrobial peptides called apidaecins that interact with PrfA and PrfB in the ribosome, preventing turnover of the termination complex (47–49). In addition, a small-scale screen of computationally selected compounds yielded molecules that bind to PrfA and PrfB and appear to inhibit release factor turnover (50). Both PrfA and PrfB are broadly conserved in bacteria, and they share the highest degree of homology of any of the proteins described here, apart from the topoisomerase subunits, making them attractive multitargets. There is one close human homolog to PrfA and PrfB, the mitochondrial translational release factor 1-like protein, that may need to be accounted for when investigating PrfA and PrfB as targets.

Group 11 is composed of Ffh and FtsY, two broadly conserved proteins involved in the cotranslational targeting of newly synthesized proteins to the bacterial inner membrane. Ffh is the protein component of the signal recognition particle (SRP), which binds to a signal sequence on nascent inner membrane proteins, while FtsY is the inner membrane receptor that binds to the SRP. Both proteins possess GTPase activity. Ffh is a proposed target of the natural product goadsporin (51), and a screen for chemical fragments binding FtsY has been undertaken (52). However, no clinical leads have been developed targeting either protein, and a dual targeting approach involving both proteins would be novel. One potential challenge with targeting Ffh and FtsY is the fairly high degree of sequence and functional homology of these proteins with the human SRP protein, SRP54, and the human SRP receptor alpha subunit.

The enzymes IspA and IspB, involved in isoprenoid biosynthesis, make up group 12. Several isoprenoid biosynthetic enzymes have been extensively studied as antibiotic targets, including undecaprenyl pyrophosphate synthase, encoded by IspU (53), and Dxr, which is the target of the antibacterial and antimalarial compound fosmidomycin (54). *ispA* encodes the enzyme farnesyl diphosphate synthase (FPPS), while *ispB* encodes octaprenyl diphosphate synthase (OPPS). Bisphosphonates drugs, used to treat osteoporosis, are inhibitors of human FPPS. Bisphosphonate compounds have been described that inhibit bacterial FPPS and OPPS (55, 56), but no compounds targeting the bacterial enzymes have progressed into the clinic. Although IspA and IspB share a reasonable degree of homology with each other, they also have homology to several human proteins, including the coenzyme Q10 biosynthetic pathway member PDSS1, implicated in inherited oxidative phosphorylation disorders (57), potentially complicating the development of multitargeting inhibitors.

Group 13 contains proteins encoded by the essential genes *dnaX* and *holB*, as well as the nonessential protein RarA. *dnaX* and *holB* encode components of the DNA polymerase III holoenzyme, the main replicative DNA polymerase in bacteria. The *dnaX*-encoded $\gamma$

and $\tau$ proteins and the *holB*-encoded $\delta'$ protein are subunits of the clamp-loader complex, which assembles the sliding clamp onto DNA, allowing for processive DNA synthesis, and also helps coordinate DNA synthesis at the leading and lagging strands (58). The $\gamma$, $\tau$, and $\delta'$ proteins have a similar ATPase core structure (27), although only the $\gamma$ and $\tau$ subunits appear to have nucleotide-binding and hydrolysis capacity. Inhibitors of core DNA polymerase III enzymes have recently shown promise (59, 60), with one inhibitor in clinical trials for treatment of *C. difficile* infection (61). However, inhibitors of the clamp loader complex have not been described. One drawback of these targets is their significant homology with subunits of the human clamp loader, replication factor C.

Group 14 contains the GTPases, Der (also called EngA) and Era, as well as the nonessential GTPase MnmE. Der and Era have GTP- and RNA-binding domains and are involved in ribosome biogenesis. Der contains two GTPase domains, both of which have homology to Era. A screen for small-molecule inhibitors of Der has been carried out (62), and a structure-based design approach for inhibitors of Der and Era has also been described (63), but no leads appear to have found in these studies. Although Der and Era are interesting targets in that ribosome biogenesis represents a potentially novel pathway for multitarget inhibition, a drawback of these targets is the existence of several small GTPases in humans with sequence and/or structural homology to the bacterial enzymes (63).

Group 15 is composed of the proteins RibD and TadA. RibD is a deaminase enzyme in the riboflavin biosynthetic pathway and has no human homologs. TadA is a tRNA adenosine deaminase that exhibits homology to human tRNA adenosine deaminase 2 (ADAT2). Interestingly, this pair represents the only group in which the individual protein members are in different COG classes. Neither RibD nor TadA appears to have any described small-molecule inhibitors, and a dual-targeting approach across different pathways would be unique.

Group 16 comprises the TsaB and TsaD proteins, which form a heterodimer in the $N^6$-L-threonylcarbamoyladenine synthase complex, responsible for the posttranscriptional modification of certain tRNAs. Although no inhibitors have been described for this complex, there is a published crystal structure of the TsaB/TsaD dimer bound to a reaction intermediate (64) that could inform inhibitor design. While TsaB does not have a close human homolog, TsaD has significant homology along its length to the human mitochondrial OSGEPL1 protein, whose loss of function has been linked with neurodegenerative disease (65).

Group 17 contains FabI and FabG, essential enzymes in the fatty acid biosynthetic pathway. Both proteins also have multiple additional hits to nonessential oxidoreductases: FabI has four additional nonessential paralogs, while FabG has 17 additional nonessential paralogs. FabI is the target of the antimicrobial drugs triclosan and isoniazid, as well as the clinical trial-stage compounds afabicin (66, 67), CG400549 (68), and MUT056399 (69). Inhibitors of FabG have also been described (70–73), but the presence of multiple isoforms of FabG in some organisms may make it an unsuitable target (71). Each protein also has several human homologs, making these targets potentially difficult for multitarget inhibitor development.

Groups 18 to 21 comprise four independent groups of tRNA synthetases. Isoleucine-tRNA ligase (IleS), a member of group 18, is the target of the topical antibiotic mupirocin. Several other tRNA synthetase inhibitors have entered clinical trials, including compounds that target methionine-tRNA ligase (74, 75) and leucine-tRNA ligase (76, 77), both also in group 18. Notably, the clinical trial of the LeuS inhibitor epetraborole was terminated after resistance developed after only 1 day of treatment (78), highlighting the need for multitargeting within this protein family. Although it is clear that tRNA synthetases have the potential to be important multitargets, the presence of close human homologs of each bacterial tRNA synthetase makes the prospects of finding a conserved drug-binding site present in bacterial tRNA synthetase paralogs but absent in the human enzymes somewhat daunting.

## DISCUSSION

Despite increasing resistance to existing antibiotics (79), novel targets have been underrepresented in recently approved antibiotics, with no novel-mechanism classes launched for Gram-negative pathogens in nearly 60 years (80). The goal of this study was to

identify potential novel multitargets for antibiotic development by identifying all essential gene products having at least one additional essential paralog in the model Gram-negative pathogen *Escherichia coli*. Using the methods presented here, 21 groups of essential paralogous proteins were identified, representing a wide range of targets in the peptidoglycan, LPS, fatty acid, isoprenoid, and riboflavin biosynthetic pathways, as well as targets related to transcription, translation, DNA replication, and membrane biogenesis.

Importantly, three groups of clinically validated multitargets were identified: the two sub-unit pairs of the DNA topoisomerases, gyrase and topo IV, and one pair of penicillin-binding proteins. In addition to providing validation of this method, the identification of gyrase, topo IV, and the PBPs FtsI and MrdA highlight the special nature of these enzyme classes as antibiotic targets. Indeed, within the last decade, at least seven new quinolone compounds have been launched, and three novel nonquinolone topoisomerase inhibitors have entered clinical trials (81). The penicillin-binding proteins also continue to be important targets in current drug discovery efforts. Combinations of $\beta$-lactam drugs with $\beta$-lactamase inhibitors represent a sizable fraction of newly approved drugs (30, 81), allowing for the continued exploitation of these multitargets while avoiding the primary mechanism of resistance to these compounds, the $\beta$-lactamase enzymes. In addition, non-$\beta$-lactam compounds that inhibit multiple PBPs are currently being explored (82), providing additional inhibitor scaffolds for this important set of targets.

**Prioritization of unexploited multitargets and examples of potential inhibitor-binding sites.** In addition to these clinically validated targets, this work uncovered 18 protein groups that are potentially promising targets for novel multitargeting antibiotics. These targets vary in the degree of homology between the paralogous partners, the similarity to their human homologs, their cellular localization, and their spectrum of conservation. Such properties, summarized in Table 2, may influence the feasibility of development of multitargeting inhibitors for these targets. For example, targets with high levels of homology between the bacterial paralogs, and/or low homology with human homologs may be the most amenable to multitarget inhibitor development. If broad-spectrum activity is desired, targets that are present in a wide range of Gram-positive and Gram-negative bacteria can be chosen for further study.

Of particular note, entry of drugs into the bacterial cytoplasm can be a formidable requirement in antibiotic drug development (83, 84), particularly for Gram-negative bacteria whose inner and outer membranes have different permeability requirements that can constrain medicinal chemistry efforts (38, 85). Thus, targets located in the outer membrane, inner membrane, or periplasmic space, such as those in groups 3 to 5, may be preferable to cytosolic targets. Finally, the use of three-dimensional protein structures in the design and optimization of inhibitors against a particular target is generally considered advantageous (86). Fortunately, structural information is available for most of the proteins described in this work (Table 1 and Table S6), indicating that structure-based inhibitor design is possible for many of the targets described here.

Perhaps the most promising investigative multitargets identified here are the FtsW and RodA enzymes of group 4. These proteins possess peptidoglycan glycosyltransferase activity and are broadly conserved in bacteria. They are located in the inner membrane, have an extensive region of homology (Fig. 2A), and do not have human homologs. RodA interacts with MrdA (PBP2, group 3) to effect side wall peptidoglycan synthesis during cell elongation (87). Similarly, FtsW acts in concert with FtsI (PBP3; group 3) to enable peptidoglycan synthesis at the cell division site (20). These parallel essential roles are reminiscent of gyrase and topo IV (groups 1 and 2), which possess similar enzymatic mechanisms but act at different points during DNA replication (88).

Inhibitors of FtsW and RodA have not yet been described, possibly because their structures and modes of action are still being fully elucidated (89–91). Interestingly, although the glycosyltransferase activity of multimodular PBPs can be inhibited by the natural product, moenomycin, neither FtsW nor RodA activity is inhibited by this compound class (18, 20).

Currently, the only experimentally derived structures of RodA/FtsW homologs are of the archaeal *Thermus thermophilus* RodA protein (90, 91), which shares 39% amino

**TABLE 2** Summary of potential antibiotic multitargets[a]

| Group no. | Protein names | Unexploited multitarget | Bacterial paralog homology[b] | Human homology[c] | Cellular localization[d] | Bacterial conservation[e] |
|---|---|---|---|---|---|---|
| 1 | GyrA/ParC | No | +++ | ++/+ | C | Broad |
| 2 | GyrB/ParE | No | +++ | ++ | C | Broad |
| 3 | FtsI/MrdA | No | +++ | None | M | G(+), partial G(−) |
| 4 | FtsW/RodA | Yes | +++ | None | M | G(+), partial G(−) |
| 5 | LolC/LolE | Yes | +++ | None | M | Partial G(−) only |
| 6 | RpoD/RpoH | Yes | ++ | None | C | G(−) only |
| 7 | DnaA/Hda | Yes | ++ | None | M+C | G(−) only |
| 8 | LpxA/LpxD | Yes | + | None | C | G(−) only |
| 9 | MurC/MurD | Yes | + | None | C | Broad |
| 10 | PrfA/PrfB | Yes | +++ | +++/++ | C | Broad |
| 11 | Ffh/FtsY | Yes | +++ | +++ | C | Broad |
| 12 | IspA/IspB | Yes | ++ | +++ | C | Broad |
| 13 | DnaX/HolB | Yes | + | ++/+ | C | Broad |
| 14 | Der/Era | Yes | + | +/++ | C | Broad |
| 15 | RibD/TadA | Yes | + | None/++ | C | G(−), partial G(+) |
| 16 | TsaB/TsaD | Yes | + | None/+++ | C | Broad |
| 17 | FabI/FabG | Yes | + | ++/+++ | C | Broad |
| 18 | ValS/IleS | Yes | +++ | +++ | C | Broad |
| 18 | ValS/LeuS | Yes | ++ | +++ | C | Broad |
| 18 | IleS/LeuS | Yes | ++ | +++ | C | Broad |
| 18 | MetG/IleS | Yes | + | +++ | C | Broad |
| 18 | MetG/LeuS | Yes | + | +++ | C | Broad |
| 19 | LysS/AspS | Yes | ++ | +++ | C | Broad |
| 19 | LysS/AsnS | Yes | + | +++ | C | G(+), partial G(−) |
| 19 | AspS/AsnS | Yes | + | +++ | C | G(+), partial G(−) |
| 20 | GltX/GlnS | Yes | ++ | +++ | C | G(−) only |
| 21 | ProS/ThrS | Yes | + | +++ | C | Broad |

[a]Qualitatively, dark gray shading indicates characteristics that are most favorable, light gray shading indicates favorable, and no shading indicates neutral or unfavorable.
[b]Homology based on E values between bacterial paralogs: +, 1e-10 ≤ E value < 0.1; ++, 1e-30 < E value < 1e-10; E value +++, ≤1e-30. Where E values for paralogs fell into different classes depending on the directionality of the search, a single E value representing the lower degree of homology is presented.
[c]Homology based on E values of each bacterial paralog with its closest human homolog: +, 1e-10 ≤ E value < 0.1; ++, 1e-30 < E value < 1e-10; E value +++, ≤1e-30; none, no detectable homology. Human homolog E values for both bacterial paralogs are represented in the order of the protein names (column 2). Where E values for the paralogs fell into the same range, a single range value is presented.
[d]C, One or both paralogs are cytoplasmic; M, both proteins are localized to the inner membrane, outer membrane, or periplasm; M+C, both proteins are localized in both the membrane and cytoplasmic compartments.
[e]Broad, both paralogs are present in both Gram-positive and Gram-negative bacteria. If either paralog has a more restricted spectrum, that spectrum is designated.

acid identity with the *E. coli* RodA protein and 34% identity with *E. coli* FtsW (Table S6). A recent report by Li et al. (89) focusing on *E. coli* FtsW shed light on the potential active site residues of this protein. These residues fall within the homologous region identified in this study and are almost all identical between *E. coli* FtsW and RodA (Fig. 2A). Mapping of these residues onto AlphaFold models of *E. coli* FtsW and RodA shows that they form a conserved cluster around a cavity on the periplasmic side of the protein (89) (Fig. 2B), a site that may be able to be targeted with FtsW/RodA multitarget inhibitors. Given the highly parallel nature of these proteins with the PBPs, their location in the inner membrane, the importance of peptidoglycan synthesis as an antibiotic target, and their broad conservation in bacteria and lack of human homologs, multitargeting inhibitors of FtsW and RodA could represent a major new avenue for antibiotic therapy.

Also, similar to the gyrase/topo IV paradigm, the highly homologous peptide release factors PrfA and PrfB (group 10, Fig. 2C) have closely related but independent functions in the cell, having different stop codon specificities. A class of antimicrobial peptides called apidaecins has been described that interact with PrfA and PrfB in the ribosome, preventing disassembly of the termination complex (47–49). Cellular effects of this interaction include accumulation of stalled ribosomes at translation termination sites, peptide release factor sequestration, and stop codon readthrough (47, 92). Structures of PrfA and PrfB in *E. coli* ribosomes (47, 93) show that these proteins share a high degree of similarity at the three-dimensional level, including in the region mediating apidaecin binding (Fig. 2D). This region also contains the conserved GGQ motif responsible for hydrolysis of the peptidyl-tRNA bond (94), allowing for release of

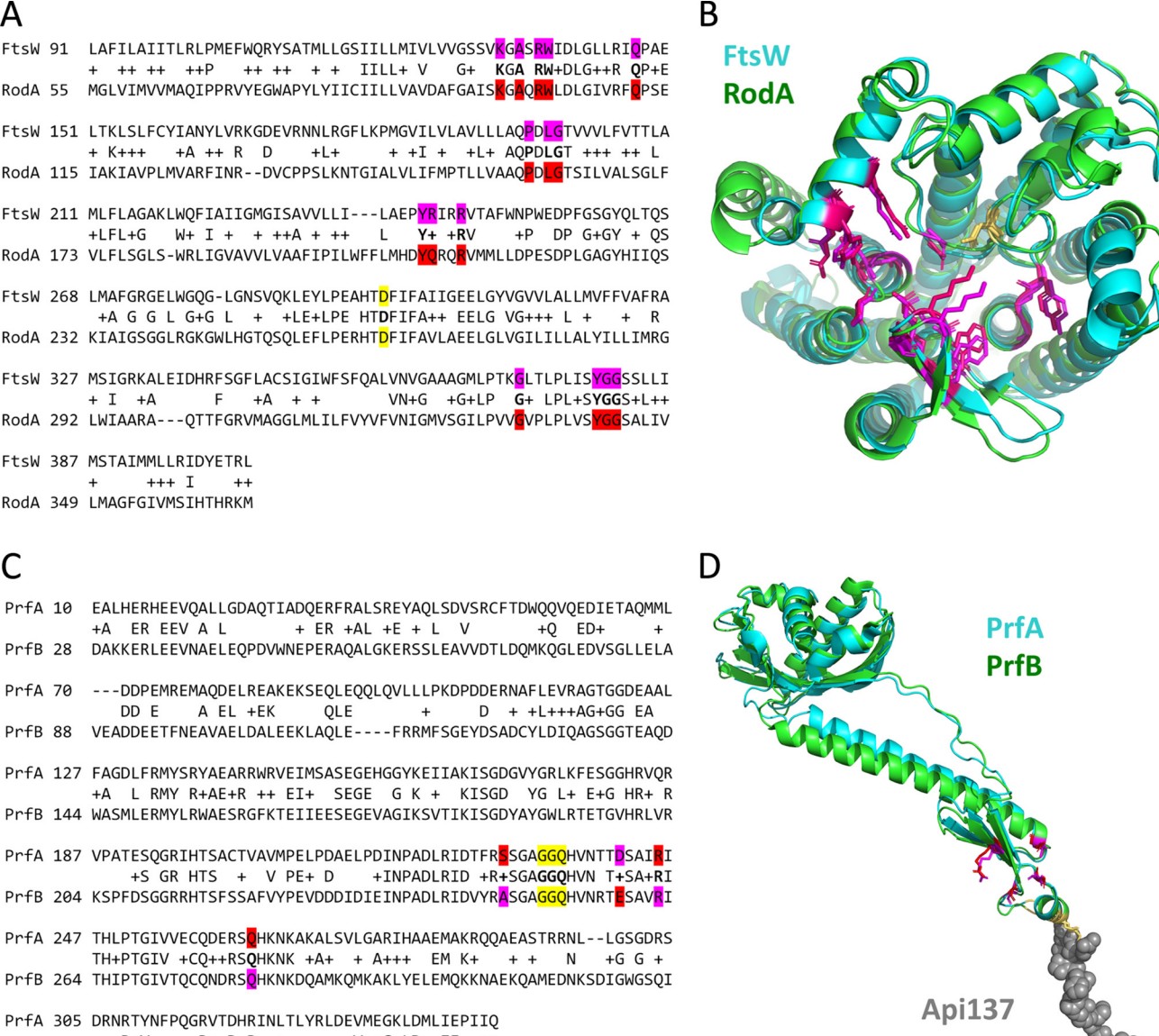

**A**

```
FtsW  91  LAFILAIITLRLPMEFWQRYSATMLLGSIILLMIVLVVGSSVKGASRWIDLGLLRIQPAE
          +  ++ ++  ++P   ++ ++  + +   IILL+ V   G+   KGA RW+DLG++R QP+E
RodA  55  MGLVIMVVMAQIPPRVYEGWAPYLYIICIILLVAVDAFGAISKGAQRWLDGIVRFQPSE

FtsW 151  LTKLSLFCYIANYLVRKGDEVRNNLRGFLKPMGVILVLAVLLLAQPDLGTVVVLFVTTLA
          + K+++    +A ++ R   D  +L+     + +I +  +L+ AQPDLGT +++ ++ L
RodA 115  IAKIAVPLMVARFINR--DVCPPSLKNTGIALVLIFMPTLLVAAQPDLGTSILVALSGLF

FtsW 211  MLFLAGAKLWQFIAIIGMGISAVVLLI---LAEPYRIRRVTAFWNPWEDPFGSGYQLTQS
          +LFL+G   W+ I +  + ++A + ++     L   Y+ +RV    +P DP G+GY + QS
RodA 173  VLFLSGLS-WRLIGVAVVLVAAFIPILWFFLMHDYQRQRVMMLLDPESDPLGAGYHIIQS

FtsW 268  LMAFGRGELWGQG-LGNSVQKLEYLPEAHTDFIFAIIGEELGYVGVVLALLMVFFVAFRA
          +A G G L G+G L +   +LE+LPE HTDFIFA++ EELG VG+++ L +   +   R
RodA 232  KIAIGSGGLRGKGWLHGTQSQLEFLPERHTDFIFAVLAEELGLVGILILLALYILLMRG

FtsW 327  MSIGRKALEIDHRFSGFLACSIGIWFSFQALVNVGAAAGMLPTKGLTLPLISYGGSSLLI
          +  I +A     F   +A + +     VN+G  +G+LP   G+ LPL+SYGGS+L++
RodA 292  LWIAARA---QTTFGRVMAGGLMLILFVYVFVNIGMVSGILPVVGVPLPLVSYGGSALIV

FtsW 387  MSTAIMMLLRIDYETRL
          +     +++ I   ++
RodA 349  LMAGFGIVMSIHTHRKM
```

**C**

```
PrfA  10  EALHERHEEVQALLGDAQTIADQERFRALSREYAQLSDVSRCFTDWQQVQEDIETAQMML
          +A ER EEV A L     + ER +AL +E + L  V     +Q ED+    +
PrfB  28  DAKKERLEEVNAELEQPDVWNEPERAQALGKERSSLEAVVDTLDQMKQGLEDVSGLLELA

PrfA  70  ---DDPEMREMAQDELREAKEKSEQLEQQLQVLLLPKDPDDERNAFLEVRAGTGGDEAAL
             DD E   A EL +EK   QLE     +    D + +L+++AG+GG EA
PrfB  88  VEADDEETFNEAVAELDALEEKLAQLE----FRRMFSGEYDSADCYLDIQAGSGGTEAQD

PrfA 127  FAGDLFRMYSRYAEARRWRVEIMSASEGEHGGYKEIIAKISGDGVYGRLKFESGGHRVQR
          +A  L RMY R+AE+R ++  EI+  SEGE  G K +  KISGD  YG L+ E+G HR+ R
PrfB 144  WASMLERMYLRWAESRGFKTEIIEESEGEVAGIKSVTIKISGDYAYGWLRTETGVHRLVR

PrfA 187  VPATESQGRIHTSACTVAVMPELPDAELPDINPADLRIDTFRSSGAGGQHVNTTDSAIRI
             +S GR HTS + V PE+ D   +INPADLRID +R+SGAGGQHVN T+SA+RI
PrfB 204  KSPFDSGGRRHTSFSSAFVYPEVDDDIDIEINPADLRIDVYRASGAGGQHVNRTESAVRI

PrfA 247  THLPTGIVVECQDERSQHKNKAKALSVLGARIHAAEMAKRQQAEASTRRNL--LGSGDRS
          TH+PTGIV +CQ++RSQHKNK +A+ + A+++ EM K+   + +   N   +G G +
PrfB 264  THIPTGIVTQCQNDRSQHKNKDQAMKQMKAKLYELEMQKKNAEKQAMEDNKSDIGWGSQI

PrfA 305  DRNRTYNFPQGRVTDHRINLTLYRLDEVMEGKLDMLIEPIIQ
          R+Y    R+ D R +     V++G LD  IE ++
PrfB 324  ---RSYVLDDSRIKDLRTGVETRNTQAVLDGSLDQFIEASLK
```

**B** FtsW RodA

**D** PrfA PrfB Api137

**FIG 2** Conserved amino acid sequences and three-dimensional structures of potential multitargets FtsW/RodA and PrfA/PrfB. (A) The region of amino acid alignment between FtsW and RodA is shown as determined by the automated BLAST search used in this study. Residues in FtsW implicated as important for catalysis (89) are highlighted in magenta, homologous residues in RodA are highlighted in red, and the putative catalytic aspartic acid residue (20, 89) for each protein is highlighted in yellow. (B) AlphaFold v2.0 (110, 111) structures of *E. coli* FtsW (AF_AFP0ABG4F1, cyan) and *E. coli* RodA (AF_AFP0ABG7F1, green) were displayed and aligned using PyMOL v2.0 (root mean square deviation [RMSD] = 1.19 Å, 1,657/2,117 atoms aligned). Catalytically important residues in FtsW (89) are shown with magenta sticks, homologous residues in RodA are indicated with red sticks, and the putative catalytic aspartic acid residue for each protein is shown with yellow sticks. An experimentally derived structure of RodA from *Thermus thermophilus* (PDB: 6BAR) (90) aligned well with the AlphaFold model of *E. coli* RodA (RMSD, 1.56 Å; 1,271/1,765 atoms aligned), indicating that the AlphaFold structures are likely to be physiologically relevant (not shown). Residues 1 to 46 of FtsW and 1 to 18 of RodA are hidden in this figure due to low model confidence in the N-terminal regions of each protein. (C) The region of amino acid alignment between PrfA and PrfB is shown as determined by the automated BLAST search used in this study. Residues that when mutated confer resistance to the apidaecin derivative Api137 (47) are highlighted in magenta, with homologous residues highlighted in red. The conserved GGQ motif in each protein is highlighted in yellow. (D) Experimentally derived structures of *E. coli* PrfA + Api137 (PDB: 5O2R, cyan, with Api137 shown as gray spheres), and *E. coli* PrfB (PDB: 5MDV, green) (93) were displayed and aligned using PyMOL v 2.0 (RMSD, 1.55 Å; 1,334/1,522 atoms aligned). Both structures are part of larger *E. coli* ribosome structures; in this figure, the rest of the ribosome is omitted from view. Residues 1 to 126 of PrfB are hidden, as the corresponding residues in PrfA are not present in the 5O2R structure. Residues involved in resistance to Api137 are indicated with magenta sticks (47), with homologous residues shown with red sticks. The catalytically important GGQ motifs in PrfA and PrfB are indicated with yellow sticks. Q252 is methylated, and the residue at position 246 is a threonine in the PrfB 5MDV structure.

the translated protein from the ribosome. Further exploration of this site or others (50) in PrfA and PrfB could yield multitargeting protein synthesis inhibitors with novel mechanisms of action, making this pair another especially exciting possible set of targets.

In summary, the essential multitargets described in this work can be prioritized based on various factors, including cellular localization of the protein targets and desired spectrum of activity. In addition, structural considerations are likely to play a key role in the design of inhibitors having balanced activity against multiple targets.

**Nonessential multitargets.** Although this work focused on essential gene products as potential multitargets, multitargeting of nonessential proteins may also hold promise. Since essentiality has mostly been determined using nonpathogenic laboratory strains of *E. coli* grown in rich media, proteins that may be necessary for growth or virulence within a host but that are not essential for growth *in vitro* are generally considered nonessential. If the absence of these activities confers a large fitness cost *in vivo*, these proteins may represent fruitful targets.

Furthermore, there is a growing appreciation that bacterial cell death upon antibiotic exposure involves important downstream effects beyond simple inhibition of the target (95–97). For example, the $\beta$-lactam mecillinam, which targets PBP2, has a lethal effect even in a cellular context in which PBP2 activity is not required for viability, by inducing a futile cycle of peptidoglycan synthesis and degradation (97). Similarly, quinolone and aminoglycoside antibiotics create toxic intermediates (DNA-protein lesions or mistranslated proteins, respectively) that have dominant negative effects on cell viability. In this light, multitargeting of any targets (essential or nonessential) whose inhibition would individually lead to a toxic cellular malfunction could require multiple mutations for drug resistance to develop, although these mutations may be more accessible in the context of nonessentiality.

Because the paralog search described here was carried out on all gene products in the pathogenic *E. coli* Sakai strain, any protein of interest can be quickly queried to check for the existence of paralogous partners. In this study, nearly half (47%, or 1,841 total proteins) of *E. coli*-conserved nonessential proteins in the *E. coli* Sakai genome had at least one conserved paralog. Thus, nonessential proteins may represent a large untapped reservoir of potential antibiotic multitargets.

**Concluding remarks.** Since the development of antibiotic resistance is an inevitable consequence of using these important drugs, there will always be a need for new antibiotics. Targeting of multiple gene products by single-agent therapeutics is a characteristic shared by many clinically successful antibiotics (6) and is likely to be an important aspect of new antibiotic classes as well.

The aim of this work was to provide a comprehensive inventory of potential protein multitargets in the bacterium *Escherichia coli* that can be used to guide antibiotic drug discovery efforts. Recent advances such as fragment-based screening (98) and DNA-encoded chemical libraries (99) that allow sampling of more chemical space than found in traditional chemical libraries, together with a better understanding of how to improve drug accumulation inside bacterial cells (100, 101), are anticipated to improve the efficiency of antibiotic lead generation. Application of these and other approaches to the multitargets described here could lead to powerful novel antibacterials with low propensities for antibiotic resistance, refilling our antibiotic arsenal for the future.

## MATERIALS AND METHODS

For this study, the *E. coli* O157:H7 strain Sakai genome was chosen for analysis because it represents a pathogenic strain of this organism and has a well-annotated genomic sequence (102). This strain was responsible for causing a significant outbreak of enterohemorrhagic illness in Japan in 1996.

Of the 5,203 protein-coding gene products annotated in the *E. coli* Sakai genome, 5,198 protein sequences (Table S1) were used in the analysis (4 were removed for having 16 or fewer amino acids, and an additional protein generated errors because it contained stop codons). Each protein sequence was imported sequentially into a Python script (EcoliProteinsBlast.py, supplemental material) that subjected it to a Biopython-based (103) BLAST search of the NCBI nonredundant (nr) database restricted to *E. coli*. The following parameters were used: "tblastn", "nr", expect=0.001, hitlist_size=20000, entrez_query="*Escherichia coli*" [organism]. tBLASTn was used for the queries rather than BLASTp to gauge conservation within *E. coli* without biasing against poorly annotated strains.

BLAST output data were stored as single files for each protein. These files were then analyzed (using EcoliParalogs.py, supplemental material) to generate a list of all the proteins and how many *E. coli* genomes

contained 1, 2, 3, 4 to 9, or 10+ high-scoring segment pairs (HSPs) for each protein sequence (Table S3). The first HSP generally corresponds to the exact or near-exact protein sequence itself being found, while additional HSPs correspond to paralogous protein sequences present within the same genomic sequence. For this work, a protein sequence was considered to have an *E. coli*-conserved match if the number of genomes hit at a particular HSP number was at least 90% of the mode number of genomic sequences hit for HSP = 1 across all gene products (mode = 1,011 sequences at the time of the analysis).

The list of essential gene products used here was compiled by defining essential genes as those found to be essential in two or more of the following studies: the Keio collection (104), the PEC database (https://shigen .nig.ac.jp/ecoli/pec/), a transposon mutagenesis study of *E. coli* ST131 (105), and a transposon mutagenesis analysis of *E. coli* K-12 (106). This resulted in a list of 313 essential genes, 309 of which were found in the *E. coli* Sakai genome (Table S2). The four missing essential proteins included three phage proteins (CohE/YmfK, DicA, RacR) and one small protein of undefined function (YceQ). All gene products aside from the 309 marked as essential were considered nonessential in this analysis.

Each essential protein having at least one additional *E. coli*-conserved paralogous match (89 total proteins) was manually annotated to determine the identity of the paralogous protein(s) and whether the paralogous partners were essential (Table S4). Essential proteins having *E. coli*-conserved essential paralogs (44 proteins) were subjected to additional manual steps, including a BLASTp (PSI-BLAST) search against the nr database restricted to both *E. coli* Sakai and human genomes to assess the relative homology of each protein for its *E. coli* paralog versus potential human homologs. E values from this analysis rather than those from the original Biopython BLAST searches are presented in Table 1 so that the bacterial and human homolog E values can be compared directly. Protein localization information for the final set of essential, paralogous proteins was obtained from EcoCyc (107) (https://ecocyc.org), and COG functional categories (108, 109) were obtained from NCBI. Conservation within other bacteria was gauged by checking for homologs of each of the proteins in a set of Gram-positive, Gram-negative, and atypical bacteria using the Database of Clusters of Orthologous Genes (https://www.ncbi.nlm.nih.gov/research/cog). See Table S5 for additional information about the bacterial conservation analysis. Figure 1 was created using SankeyMATIC (https://sankeymatic.com).

## SUPPLEMENTAL MATERIAL

Supplemental material is available online only.
**SUPPLEMENTAL FILE 1**, XLSX file, 1 MB.
**SUPPLEMENTAL FILE 2**, PDF file, 0.8 MB.
**SUPPLEMENTAL FILE 3**, XLSX file, 1.7 MB.

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
