## [Reviewer comments · Microbiology Spectrum]

Microbiology Spectrum

Essential Paralogous Proteins as Potential Antibiotic Multi-Targets in *Escherichia coli*

Christine Hardy

Corresponding Author(s): Christine Hardy, CDH Consulting

Review Timeline:

Submission Date:	June 18, 2022
Editorial Decision:	July 13, 2022
Revision Received:	September 13, 2022
Editorial Decision:	October 3, 2022
Revision Received:	October 29, 2022
Accepted:	November 1, 2022

Editor: Silvia Cardona

Reviewer(s): Disclosure of reviewer identity is with reference to reviewer comments included in decision letter(s). The following individuals involved in review of your submission have agreed to reveal their identity: Francis E Nano (Reviewer #3)

Transaction Report:

DOI: <https://doi.org/10.1128/spectrum.02043-22>

July 13, 2022

Dr. Christine Diane Hardy
CDH Consulting
4 Bitterwood
Irvine, CA

Re: Spectrum02043-22 (Essential Paralogous Proteins as Potential Antibiotic Multi-Targets in *Escherichia coli*)

Dear Dr. Christine Diane Hardy:

Thank you for submitting your manuscript to Microbiology Spectrum. Three expert reviewers submitted their comments regarding your article. As you will see reviewers found your work relevant to the field of antibiotic discovery. However, reviewers found that your work is too preliminary for the article to be accepted in its present form.

I agree with the reviewers that a more in-depth analysis of the targets identified in groups 10-21 would strengthen this work substantially. For example, you could look at structural analysis of the targets to identify possible drug binding sites. You could look at reports of synthetic lethal interactions to validate target pairs.

Second, I suggest improving how the data is shown. Most of your data remain in raw tables, increasing the perception that your work is mostly preliminary. If you could analyze some of the targets proposed more thoroughly with the aid of figures, your article could improve substantially.

Finally, I would suggest clarifying the concept of essential genes and essential proteins in the context of the existence of paralogs. Specifically, how can a gene be essential (no mutant can be obtained) if there are paralog genes whose products may perform the same function? if they do not perform the same function, then a drug that targets the essential protein will produce a lethal phenotype regardless of the presence of a paralog. If the essential gene is mutated to generate resistance, the presence of the paralog will have no role.

When submitting the revised version of your paper, please provide (1) point-by-point responses to the issues raised by the reviewers as file type "Response to Reviewers," (2) comment on my points in your cover letter, and (3) a PDF file that indicates the changes from the original submission (by highlighting or underlining the changes) as file type "Marked Up Manuscript - For Review Only". Please use this link to submit your revised manuscript - we strongly recommend that you submit your paper within the next 60 days or reach out to me. Detailed instructions on submitting your revised paper are below.

Link Not Available

Sincerely,

Silvia Cardona

Journals Department
American Society for Microbiology
1752 N St., NW

Reviewer comments:

Reviewer #1 (Comments for the Author):

Overall, the manuscript highlights the success and potential of taking a "one drug, multiple target" approach to drug discovery. Specifically, it describes essential *E. coli* paralogs as potential drug targets. A bioinformatic approach was well-defined that resulted in the classification of essential and conserved *E. coli* genes into 21 groups consisting of dual to multi (3+) targets consisting of paralogous proteins. The bioinformatic approach was, expectedly, supported by known cases of clinically relevant drugs and other early-stage inhibitors and or molecular probes. Accordingly, the article provides a brief review of known systems before presenting a set of potential new drug targets in *E. coli*.

Specific Comments and Suggestions

Since many of the identified groups (1-9) have been previously described by others, this article acts as a partial review of those cases. It is my opinion that a more in-depth review of those may better serve interested readers on the overarching theme of antibiotics with multiple targets. The description of the potential targets is cursory at best: protein functions are presented, however, many details are missing that would speak directly to the potential of developing a single compound for the inhibition of more than one homologous protein. For example, a recent report on FtsW (PLoS Genetics. 2022. 18:e1009993) has helped to define catalytic residues. How do these compare to those in the sequence of RodA?

Groups 10-21 appear to represent the novel findings in the report, though they raise issues that are related to the comment above: divergence between targets. In the context of the members that belong to these groups of paralogs, divergence from a human homolog is invoked as an argument for the *E. coli* proteins as reasonable targets. In contrast, similarity (e.g. Ffh, FtsY and SRP54) prevents antibiotic development in some cases. How does the balance of similarity and divergence compare to that within the groups of paralogs?

The groups are thematically separated; however, some groups in 10-21 have small molecule/peptide interacting partners.

On Lines 55-57, resistance to multitarget drugs is said to "require that all genes involved mutate," which is misleading. Technically, the sentence begins by claiming this for "target-related resistance," however, mutations that results in varied cell permeability or membrane character, including mutations to pathways that are not directly targeted, for example, result in AMR to multitarget drugs.

Finally, a discussion in the context of other known resistance mechanisms, for example, ribosomal mutations, would be helpful. In this case, non-equivalent ribosomes generated by SNPs that result in AMR are known. Accordingly, what is known about the clinically used drugs' mechanisms of resistance. Can one paralog provide resistance to another? Are gene duplications events known? Along these lines, there are known bacterial systems - antibiotic biosynthetic gene clusters - that encode for non-functional homologs of their targets. In fact, identifying these proteins has been used as a strategy to identify targets of new compounds (see ACS Chem. Biol. 2015, 10, 12, 2841-2849).

Reviewer #2 (Comments for the Author):

This manuscript fills a void in the literature - as the author says "Although families of paralogous proteins were noted soon after the publication of the first bacterial genomes, a genomic-scale description of paralogous essential proteins that could be investigated for antibiotic multi-targeting has not been reported." The author accepts and promulgates the idea that antibacterial agents that have multiple targets should have a lower potential for target-based resistance than single-targeted agents, which I agree with. The idea of multitargeting has been expressed before and this work should provide a useful set of potential targets. The author carried out a well-planned analysis of all the paralogous proteins of a standard strain of *E. coli* - arriving at several groups of paralogs, focusing on those which are essential and well-conserved among *E. coli* and with low human homology.

Most of the comments below are general thoughts that the author could consider. Only a few should require checking or modification (as noted).

The definition of what constitutes a good target (or target pair) is OK - but should really include the target location - as, especially in Gram-negatives, cytoplasmic targets will be problematic. This is noted in the Discussion, but should also be brought up in the Introduction as well.

The finding of Groups 1-3 is not surprising, as their targeting was the basis for the multitarget hypothesis. But their identification by this methodology, does indeed verify this. The beta-lactams also target PBP1a and 1b (MrcA and MrcB) - but those do

not appear essential because they are each but not both dispensable [Kato, et al 1985 doi:10.1007/bf00425435 (1985)]. Binding to PBP1a or 1b is important for the efficacy of these drugs. This points up one problem with defining "essential" genes - the activity of these proteins is essential. Such pairs, of course, should not be considered as multitargets.

Groups 4-9 appear to be the best candidate multitargets. It is quite surprising that no inhibitors for FtsW and RodA have been found. While the transglycosylase activities for each have been relatively recently discovered, inhibitors of peptidoglycan synthesis have been sought by empirical screening methods for >80 years, mostly among natural products. These should definitely be pursued. LolC and E (group 5) are also paralogs of interest and are well-located.

Groups 6-9, however, are cytoplasmically located. While there are cytoplasmically targeted antibacterials, they (or their progenitors) were generally discovered by their whole cell activity, rather than by biochemical assay. The transcription factors, sigma-70 and sigma-32, are certainly apt targets, aside from their location. While DnaA has been screened for (Fossum, S. et al. doi:<https://doi.org/10.1111/j.1574-6968.2008.01103.x> (2008).) no inhibitors have been reported. The pairing with Hda is, as the author notes, possibly problematic- especially since it has less of a spectrum. Also, DnaA mutants are suppressed by a variety of mutations in other genes, some of which appear to supply alternate initiation methods - so DnaA has always been a somewhat questionable target. Once again, essentiality may be "flexible" if suppressors can arise at high frequency. The lipidA pathway targets LpxA and LpxD are of interest, but cytoplasmic location is likely problematic. The Mur ligases, Mur C, D, E and F have been repeatedly targeted with little success for a whole cell active compound. The phosphate-rich intermediates may be poor templates for permeable compounds (or require suitable phosphate isosteres).

The discussion of essential multitargets is very good. The author makes several arguments for the possibility of non-essential targets. As to non-essential targets, lines 424-426 propose that inhibition of non-essential targets may lead to toxic events downstream. However, this could use a more detailed explanation, a possible example. Also, downstream toxic events would likely select for resistance in downstream genes. Other points the author makes- the differences among organisms, ignorance of the "medium" in the host, etc. are valuable. Another point about non-essential targets in multitargeting: Since the notion that multitargeting can reduce resistance due to the necessity for multiple mutations (in the multiple targets), the degree of that reduction will rely on the frequency of viable mutations in those targets. Mutations in essential genes can lead to lethality, while in non-essential genes, they will (most likely) be viable. This should lead to a higher baseline (before drug challenge) of "resistance" mutations in non-essential genes, thus lowering the barrier to obtaining resistance mutations in both targets. In general, this paper is well written, contains useful information and should be of interest to all those interested in antibacterial discovery.

Specific Comments

Line 181 Isn't it GyrB and ParE that are paralogs?

Reviewer #3 (Comments for the Author):

In this work the author posits that the identification of paralogues of essential bacterial proteins, which are themselves essential, will allow the development of antimicrobials that target two or more essential proteins; by having multi-targeted antimicrobials the development of resistance will be slowed. The author uses *E. coli* as a model system and performed bioinformatics analysis that identified conserved paralogues of essential protein. Of the 309 essential proteins that were conserved in all of the *E. coli* strains examined 89 had conserved *E. coli* paralogues; 44 of these had *E. coli* paralogues that are essential. The author divided these 89 into 21 protein groups of related function. Importantly two groups, the topoisomerases and the penicillin binding protein groups represent groups for which multi-targeted antibiotics already exist in clinical use, adding credence to the driving hypothesis.

The manuscript is well written and well-organized, and presents a reasonably thorough discussion of the different essential gene groups that have essential paralogues. The central idea is one that addresses an extraordinarily important global health issue.

A weakness of the work is its preliminary findings. There are no biochemical experiments to support the central hypothesis and the author relies on the past development of the fluoroquinolones and β -lactams as evidence of the validity of her idea. If the manuscript is considered a pure bioinformatics analysis then further work could have examined the three dimensional structure of essential paralogues if only in domains that are highly conserved. I don't know if it's quite appropriate yet, but at some point reviewers will expect an AlphaFold analysis of protein structures central to a bioinformatics manuscript.

Staff Comments:

Preparing Revision Guidelines

To submit your modified manuscript, log onto the eJP submission site at <https://spectrum.msubmit.net/cgi-bin/main.plex>. Go to

Author Tasks and click the appropriate manuscript title to begin the revision process. The information that you entered when you first submitted the paper will be displayed. Please update the information as necessary. Here are a few examples of required updates that authors must address:

Please return the manuscript within 60 days; if you cannot complete the modification within this time period, please contact me. If you do not wish to modify the manuscript and prefer to submit it to another journal, please notify me of your decision immediately so that the manuscript may be formally withdrawn from consideration by Microbiology Spectrum.

Response to reviewers

Microbiology Spectrum Submission: Spectrum02043-22

Manuscript title: Essential Paralogous Proteins as Potential Antibiotic Multi-Targets in *Escherichia coli*

Author: Christine Hardy

September 10, 2022

Reviewer #1 (Comments for the Author):

Overall, the manuscript highlights the success and potential of taking a "one drug, multiple target" approach to drug discovery. Specifically, it describes essential *E. coli* paralogs as potential drug targets. A bioinformatic approach was well-defined that resulted in the classification of essential and conserved *E. coli* genes into 21 groups consisting of dual to multi (3+) targets consisting of paralogous proteins. The bioinformatic approach was, expectedly, supported by known cases of clinically relevant drugs and other early-stage inhibitors and or molecular probes. Accordingly, the article provides a brief review of known systems before presenting a set of potential new drug targets in *E. coli*.

Author response: This is a great summary – thank you for reading the manuscript so carefully!

Specific Comments and Suggestions

Since many of the identified groups (1-9) have been previously described by others, this article acts as a partial review of those cases. It is my opinion that a more in-depth review of those may better serve interested readers on the overarching theme of antibiotics with multiple targets. The description of the potential targets is cursory at best: protein functions are presented, however, many details are missing that would speak directly to the potential of developing a single compound for the inhibition of more than one homologous protein. For example, a recent report on FtsW (PLoS Genetics. 2022. 18:e1009993) has helped to define catalytic residues. How do these compare to those in the sequence of RodA?

*Author response: I agree that the discussion of Groups 1-9 is a largely a review of these targets and that some of these targets have been described as multi-targets by others. However, this work is not intended to be just a review of previously described targets. Rather it is intended to answer the question: what are all such multi-targets in *E. coli* that we can identify bioinformatically? Given that there are 21 protein groups with 44 proteins total, it would be difficult for me to describe each target in much more detail. I have included as many references as possible to indicate prior work that has been carried out on these targets which can serve as a reference for readers wanting additional information.*

In this resubmission, I have included a new column in Table 1 listing available protein structures for each of the targets and included a new Supplemental Table (Table S6) detailing available structures of both the bacterial targets and their human homologs. I have also expanded the discussion about FtsW and RodA to include the potential for inhibition of these enzymes and included the reference you mention - see lines 432-438. Thank you for pointing out that reference – I think the data support the idea that these proteins are promising multi-targets!

Groups 10-21 appear to represent the novel findings in the report, though they raise issues that are related to the comment above: divergence between targets. In the context of the members that belong to these groups of paralogs, divergence from a human homolog is invoked as an argument for the *E. coli*

proteins as reasonable targets. In contrast, similarity (e.g. Ffh, FtsY and SRP54) prevents antibiotic development in some cases. How does the balance of similarity and divergence compare to that within the groups of paralogs?

Author response: I agree that this aspect of how close the paralogs are to each other versus how close the human homolog(s) are was not addressed in a systematic way in the original submission. I have now added an overview table (Table 2) that I believe allows more direct comparison of the relative e-values in a more compact form. I also discuss this briefly in the new "Prioritization of unexploited multitargets" section.

The groups are thematically separated; however, some groups in 10-21 have small molecule/peptide interacting partners.

Author response: I have indicated which proteins have catalytic activity, and which have known inhibitors in Table 1. These include both small molecule and peptide inhibitors.

On Lines 55-57, resistance to multitarget drugs is said to "require that all genes involved mutate," which is misleading. Technically, the sentence begins by claiming this for "target-related resistance," however, mutations that results in varied cell permeability or membrane character, including mutations to pathways that are not directly targeted, for example, result in AMR to multitarget drugs.

Finally, a discussion in the context of other known resistance mechanisms, for example, ribosomal mutations, would be helpful. In this case, non-equivalent ribosomes generated by SNPs that result in AMR are known. Accordingly, what is known about the clinically used drugs' mechanisms of resistance. Can one paralog provide resistance to another? Are gene duplications events known? Along these lines, there are known bacterial systems - antibiotic biosynthetic gene clusters - that encode for non-functional homologs of their targets. In fact, identifying these proteins has been used as a strategy to identify targets of new compounds (see ACS Chem. Biol. 2015, 10, 12, 2841-2849).

Author response: I have now greatly expanded the introductory discussion of AMR to include non-target AMR such as cell permeability/efflux, antibiotic inactivation, and the presence of alternative drug-resistant targets - see lines 63-72. These mechanisms are relevant to the action of beta-lactams, quinolones, and triclosan, clinically used drugs that are discussed in the manuscript.

Based on the high but non-identical nature of the paralogs, I do not believe the paralogs can provide resistance to one another, say through a recombination event as is possible for the identical or nearly identical ribosomal rRNA genes. As far as I know, that has not been observed with the topoisomerases, and they share the highest degree of homology of all the multi-targets identified in this study.

Finally, although the article you mention is very interesting and I enjoyed reading it, I don't believe most clinical isolates contain such biosynthetic gene clusters. Thank you for bringing this issue to my attention though. I didn't know about those novo^R gyrB proteins encoded by novo-producing strains - super interesting! I sincerely hope I have adequately addressed your concerns about the resistance mechanisms discussed in the manuscript.

Reviewer #2 (Comments for the Author):

This manuscript fills a void in the literature - as the author says "Although families of paralogous proteins were noted soon after the publication of the first bacterial genomes, a genomic-scale description of paralogous essential proteins that could be investigated for antibiotic multi-targeting has not been reported." The author accepts and promulgates the idea that antibacterial agents that have multiple

targets should have a lower potential for target-based resistance than single-targeted agents, which I agree with. The idea of multitargeting has been expressed before and this work should provide a useful set of potential targets. The author carried out a well-planned analysis of all the paralogous proteins of a standard strain of E.coli - arriving at several groups of paralogs, focusing on those which are essential and well-conserved among E.coli and with low human homology.

Author response: Thank you for this very nice summary! I'm glad you found this work of interest.

Most of the comments below are general thoughts that the author could consider. Only a few should require checking or modification (as noted).

The definition of what constitutes a good target (or target pair) is OK - but should really include the target location - as, especially in Gram-negatives, cytoplasmic targets will be problematic. This is noted in the Discussion, but should also be brought up in the Introduction as well.

Author response: I agree, this is a very important point and has represented a true bottleneck in antibiotic discovery. In this resubmission, I have included a new summary table, Table 2, that makes it very clear that only 3 groups contain only non-cytoplasmic targets, and I have emphasized it more in the new "Prioritization of unexploited multi-targets" section of the Discussion – see lines 410-414.

I feel somewhat uncomfortable presenting cytoplasmic localization in the Introduction since there are examples of cytoplasmically located Gram (-) agents (e.g. the quinolones), and there is promising work being done in this area - see for example, Richter et al, Nature 545, 299–304 (2017). I feel including this point in the Discussion allows the reader to make their own decision regarding prioritization of targets without being biased up front.

The finding of Groups 1-3 is not surprising, as their targeting was the basis for the multitarget hypothesis. But their identification by this methodology, does indeed verify this. The beta-lactams also target PBP1a and 1b (MrcA and MrcB) - but those do not appear essential because they are each but not both dispensible [Kato, et al 1985 doi:10.1007/bf00425435 (1985)]. Binding to PBP1a or 1b is important for the efficacy of these drugs. This points up one problem with defining "essential" genes - the activity of these proteins is essential. Such pairs, of course, should not be considered as multitargets.

Author response: I have tried to describe more clearly what an essential paralogous protein is in lines 84-86 of the marked-up manuscript. I agree that those non-essential PBPs are important targets and they are mentioned in lines 217-220.

Groups 4-9 appear to be the best candidate multitargets. It is quite surprising that no inhibitors for FtsW and RodA have been found. While the transglycosylase activities for each have been relatively recently discovered, inhibitors of peptidoglycan synthesis have been sought by empirical screening methods for >80 years, mostly among natural products. These should definitely be pursued. LolC and E (group 5) are also paralogs of interest and are well-located.

Groups 6-9, however, are cytoplasmically located. While there are cytoplasmically targeted antibacterials, they (or their progenitors) were generally discovered by their whole cell activity, rather than by biochemical assay. The transcription factors, sigma-70 and sigma-32, are certainly apt targets, aside from their location. While DnaA has been screened for (Fossum, S. et al. doi:<https://doi.org/10.1111/j.1574-6968.2008.01103.x> (2008).) no inhibitors have been reported. The pairing with Hda is, as the author notes, possibly problematic- especially since it has less of a spectrum. Also, DnaA mutants are suppressed by a variety of mutations in other genes, some of which appear to supply alternate initiation methods - so DnaA has always been a somewhat questionable target. Once again, essentiality may be "flexible" if suppressors can arise at high frequency. The lipidA pathway targets LpxA and LpxD are of interest, but cytoplasmic location is likely problematic. The Mur ligases, Mur C, D, E and F have been

repeatedly targeted with little success for a whole cell active compound. The phosphate-rich intermediates may be poor templates for permeable compounds (or require suitable phosphate isosteres).

Author response: I agree with everything above. Again, I hope the new summary table (Table 2) makes it clear that only Groups 3-5 contain all non-cytoplasmic targets.

The discussion of essential multitargets is very good. The author makes several arguments for the possibility of non-essential targets. As to non-essential targets, lines 424-426 propose that inhibition of non-essential targets may lead to toxic events downstream. However, this could use a more detailed explanation, a possible example. Also, downstream toxic events would likely select for resistance in downstream genes. Other points the author makes- the differences among organisms, ignorance of the "medium" in the host, etc. are valuable. Another point about non-essential targets in multitargeting: Since the notion that multitargeting can reduce resistance due to the necessity for multiple mutations (in the multiple targets), the degree of that reduction will rely on the frequency of viable mutations in those targets. Mutations in essential genes can lead to lethality, while in non-essential genes, they will (most likely) be viable. This should lead to a higher baseline (before drug challenge) of "resistance" mutations in non-essential genes, thus lowering the barrier to obtaining resistance mutations in both targets.

Author response: In the resubmission, I have fleshed out the non-essential paralog section a bit more and included an example of this type of toxic malfunctioning effect – see lines 459-463. Perhaps it would be easier to get mutations in non-essential targets since a larger mutational landscape is possible. Rather than going into this in detail in the manuscript, I have changed "would" to "could" in line 464.

In general, this paper is well written, contains useful information and should be of interest to all those interested in antibacterial discovery.

Specific Comments

Line 181 Isn't it GyrB and ParE that are paralogs?

Author response: Yes, fixed. Thank you for catching that error.

Reviewer #3 (Comments for the Author):

In this work the author posits that the identification of paralogues of essential bacterial proteins, which are themselves essential, will allow the development of antimicrobials that target two or more essential proteins; by having multi-targeted antimicrobials the development of resistance will be slowed. The author uses E. coli as a model system and performed bioinformatics analysis that identified conserved paralogues of essential protein. Of the 309 essential proteins that were conserved in all of the E. coli strains examined 89 had conserved E. coli paralogues; 44 of these had E. coli paralogues that are essential. The author divided these 89 into 21 protein groups of related function. Importantly two groups, the topoisomerases and the penicillin binding protein groups represent groups for which multi-targeted antibiotics already exist in clinical use, adding credence to the driving hypothesis.

The manuscript is well written and well-organized, and presents a reasonably thorough discussion of the different essential gene groups that have essential paralogues. The central idea is one that addresses an extraordinarily important global health issue.

Author response: Thank you for this thorough summary! I truly did carry out this study with the goal of helping to address the AMR crisis.

A weakness of the work is its preliminary findings. There are no biochemical experiments to support the central hypothesis and the author relies on the past development of the fluoroquinolones and β -lactams as evidence of the validity of her idea. If the manuscript is considered a pure bioinformatics analysis then further work could have examined the three dimensional structure of essential paralogues if only in domains that are highly conserved. I don't know if it's quite appropriate yet, but at some point reviewers will expect an AlphaFold analysis of protein structures central to a bioinformatics manuscript.

Author response: This work is indeed intended to be a fully bioinformatic analysis. While I agree that a detailed structural analysis of the targets would be very insightful, I feel it is beyond the scope of the current work and represents an excellent follow-up study that is better carried out by a bona fide structural biologist who could assess the similarity of potential drug binding pockets and other structural characteristics that might indicate favorable or unfavorable target druggability.

To address your concern, and to provide additional information for readers who might be interested, I have now added a new column in Table 1 that includes PDB codes of representative protein structures for each of the protein targets, and I provide additional information about available structures in a new supplemental table (Table S6). In my opinion, this information adds quite a bit of information to the current work even if each structure is not discussed in detail, because it shows that the 3-D structure of the vast majority of these proteins is known and this information is available to drug development researchers. It also indicates that most of these targets are amenable to crystallography (or EM-based structural methods) so researchers could potentially set up their own structural studies evaluating the binding of potential inhibitors.

The availability of this structural data is now discussed in a new section entitled "Prioritization of unexploited multi-targets" within the Discussion – see lines 417-420. I hope this new material addresses your concerns.

Response to Editor's comments

Thank you for submitting your manuscript to Microbiology Spectrum. Three expert reviewers submitted their comments regarding your article. As you will see reviewers found your work relevant to the field of antibiotic discovery. However, reviewers found that your work is too preliminary for the article to be accepted in its present form.

I agree with the reviewers that a more in-depth analysis of the targets identified in groups 10-21 would strengthen this work substantially. For example, you could look at structural analysis of the targets to identify possible drug binding sites. You could look at reports of synthetic lethal interactions to validate target pairs.

Second, I suggest improving how the data is shown. Most of your data remain in raw tables, increasing the perception that your work is mostly preliminary. If you could analyze some of the targets proposed more thoroughly with the aid of figures, your article could improve substantially.

Finally, I would suggest clarifying the concept of essential genes and essential proteins in the context of the existence of paralogs. Specifically, how can a gene be essential (no mutant can be obtained) if there are paralog genes whose products may perform the same function? if they do not perform the same function, then a drug that targets the essential protein will produce a lethal phenotype regardless of the presence of a paralog. If the essential gene is mutated to generate resistance, the presence of the paralog will have no role.

Author response: I appreciate all the reviewer comments and your comments above and have attempted to address them in this resubmission.

First, to address the issue of the work seeming "too preliminary," I have added a summary table (Table 2), that allows the reader to quickly appreciate the findings of this work. Now, the data are not just presented in the larger "raw table"-style Table 1 but can be accessed in a summarized format. I have also created a new section in the Discussion entitled "Prioritization of unexploited multi-targets" that I hope demonstrates that the data have been extensively analyzed and conclusions can be drawn about the relative merit of the targets.

Second, to get at the issue of structural analysis of the target proteins, I have added a column in Table 1 listing representative protein structures available in the PDB, and I have added another Supplemental Table, Table S6, that goes into more detail about the available protein structures. I have also now mentioned the availability of the structural data in the new "Prioritization of unexploited multi-targets" section of the Discussion. I believe that going into detail about each potential target's protein structure is beyond the scope of this work. However, I feel in referencing this additional structural information, I have pointed out the availability of such data and I hope that researchers more experienced in structural biology than me will analyze it in depth to determine which targets are most amenable to inhibition.

Finally, I have clarified several points throughout the manuscript indicated by the reviewers as needing improvement including the concept of paralogous proteins having different essential roles in the cell that you mentioned (see lines 84-86 in the Introduction of the marked-up manuscript).

October 3, 2022

Dr. Christine Diane Hardy
CDH Consulting
4 Bitterwood
Irvine, CA

Re: Spectrum02043-22R1 (Essential Paralogous Proteins as Potential Antibiotic Multi-Targets in *Escherichia coli*)

Dear Dr. Christine Diane Hardy:

Thank you for submitting your manuscript to Microbiology Spectrum. As you will see from their comments, reviewers were somehow positive about your work. One reviewer pointed out that your work would highlight potential targets for antibiotic drug discovery. However, one reviewer was disappointed that a structural analysis of some of the targets was not shown. In agreement with that reviewer, I think you could at least show a figure comparing the structures of two paralog targets. For example, take your description of the most promising pair of targets RodA and FtsW. You could show a figure with a pairwise alignment of both proteins, showing regions of similarity, the substrate binding sites and a cartoon of the structures. Figures attract the attention of the reader towards a point of interest. I strongly recommend you include at least one figure to make your point.

If you decide to submit the revised version of your paper with my recommendation, please provide (1) point-by-point responses to the issues raised by the reviewers as file type "Response to Reviewers," not in your cover letter, and (2) a PDF file that indicates the changes from the original submission (by highlighting or underlining the changes) as file type "Marked Up Manuscript - For Review Only". Please use this link to submit your revised manuscript - we strongly recommend that you submit your paper within the next 60 days or reach out to me. Detailed instructions on submitting your revised paper are below.

Link Not Available

Sincerely,

Silvia Cardona

Journals Department
Reviewer comments:

Reviewer #2 (Comments for the Author):

I reviewed a previous version of this paper and had a positive response. Unlike the other reviewers, I did not feel that this was "preliminary: - as the aim was to provide a bioinformatic analysis of possible paralog target for novel antibacterial agents, which it did. In this revised version, the author has responded well to my few comments but also to the concerns of the others by adding information to the tables (especially references to structural information) and addressing some underanalyzed sections. I reiterate my approbation. This work provides a good background for others to proceed with drug discovery using this information.

Reviewer #3 (Comments for the Author):

The revised manuscript is, like the original, well-written and well-organized. I am not going to belabor the point, but I still believe the author could have included some structural analysis, and perhaps bringing on a new author to accomplish this might have been wise. Even without a collaborator it should have been straight forward to determine if the regions of high homology are associated with ligand interaction and thus relevant to developing inhibitors.

Table 2, column 3: The term "Unexploited multi-target" gets confusing when the answer is "no." You end up with a double negative. Consider using something like "Currently exploited target."

Staff Comments:

Preparing Revision Guidelines

Please return the manuscript within 60 days; if you cannot complete the modification within this time period, please contact me. If you do not wish to modify the manuscript and prefer to submit it to another journal, please notify me of your decision immediately so that the manuscript may be formally withdrawn from consideration by Microbiology Spectrum.

Response to reviewers

Microbiology Spectrum Submission: Spectrum02043-22R1

Manuscript title: Essential Paralogous Proteins as Potential Antibiotic Multi-Targets in *Escherichia coli*

Author: Christine Hardy

October 28, 2022

Reviewer #2 (Comments for the Author):

I reviewed a previous version of this paper and had a positive response. Unlike the other reviewers, I did not feel that this was "preliminary: - as the aim was to provide a bioinformatic analysis of possible paralog target for novel antibacterial agents, which it did. In this revised version, the author has responded well to my few comments but also to the concerns of the others by adding information to the tables (especially references to structural information) and addressing some underanalyzed sections.

I reiterate my approbation. This work provides a good background for others to proceed with drug discovery using this information.

Author response: Thank you for your continued support in the publication of this work.

Reviewer #3 (Comments for the Author):

The revised manuscript is, like the original, well-written and well-organized. I am not going to belabor the point, but I still believe the author could have included some structural analysis, and perhaps bringing on a new author to accomplish this might have been wise. Even without a collaborator it should have been straight forward to determine if the regions of high homology are associated with ligand interaction and thus relevant to developing inhibitors.

Author response: Thank you for reviewing the manuscript again. In this second revision, I have included structural analyses of two sets of potential multi-targets: FtsW/RodA and PrfA/PrfB – see Figure 2. I have also included information about potential inhibitor binding sites in these targets (see lines 321-359 in the marked-up manuscript.) From this analysis, it is clear that the regions of homology found in the original analysis correspond to regions of three-dimensional similarity and include potential inhibitor binding sites. I sincerely hope this new analysis addresses your concerns.

Table 2, column 3: The term "Unexploited multi-target" gets confusing when the answer is "no." You end up with a double negative. Consider using something like "Currently exploited target."

Author response: While I agree that this term does bring on a double negative, I'd rather emphasize the new multi-targets with a "Yes" than a "No", and I feel uncomfortable calling them novel multi-targets as some have been described before. Unless you feel strongly about this, I'd prefer to leave it as is.

November 1, 2022

Dr. Christine Diane Hardy
CDH Consulting
4 Bitterwood
Irvine, CA

Re: Spectrum02043-22R2 (Essential Paralogous Proteins as Potential Antibiotic Multi-Targets in *Escherichia coli*)

Dear Dr. Christine Diane Hardy:

Your manuscript has been accepted, and I am forwarding it to the ASM Journals Department for publication. You will be notified when your proofs are ready to be viewed.

Sincerely,

Silvia Cardona
Editor, Microbiology Spectrum

Journals Department
Supplemental Tables S1-S6: Accept
Supplemental Material Legends and Supplemental Text S1: Accept
Supplemental Dataset S1: Accept